# Manifold learning analysis suggests strategies to align single-cell multimodal data of neuronal electrophysiology and transcriptomics

Jiawei Huang[1,5], Jie Sheng[2] & Daifeng Wang [2,3,4✉]

Recent single-cell multimodal data reveal multi-scale characteristics of single cells, such as transcriptomics, morphology, and electrophysiology. However, integrating and analyzing such multimodal data to deeper understand functional genomics and gene regulation in various cellular characteristics remains elusive. To address this, we applied and benchmarked multiple machine learning methods to align gene expression and electrophysiological data of single neuronal cells in the mouse brain from the Brain Initiative. We found that nonlinear manifold learning outperforms other methods. After manifold alignment, the cells form clusters highly corresponding to transcriptomic and morphological cell types, suggesting a strong nonlinear relationship between gene expression and electrophysiology at the cell-type level. Also, the electrophysiological features are highly predictable by gene expression on the latent space from manifold alignment. The aligned cells further show continuous changes of electrophysiological features, implying cross-cluster gene expression transitions. Functional enrichment and gene regulatory network analyses for those cell clusters revealed potential genome functions and molecular mechanisms from gene expression to neuronal electrophysiology.

[1] Department of Statistics, University of Wisconsin - Madison, Madison, WI 53706, USA. [2] Waisman Center, University of Wisconsin – Madison, Madison, WI 53705, USA. [3] Department of Biostatistics and Medical Informatics, University of Wisconsin – Madison, Madison, WI 53706, USA. [4] Department of Computer Sciences, University of Wisconsin – Madison, Madison, WI 53706, USA. [5] Present address: Carl H. Lindner College of Business, University of Cincinnati, Cincinnati, OH 45223, USA. ✉email: daifeng.wang@wisc.edu

Recent single-cell technologies have generated great excitement and interest in studying functional genomics at cellular resolution[1]. For example, recent Patch-seq techniques enable measuring multiple characteristics of individual neuronal cells, including transcriptomics, morphology, and electrophysiology in the complex brains, also known as single-cell multimodal data[2]. Further computational analyses have clustered cells into many cell types for each modality. The same type's cells share similar characteristics: t-type by transcriptomics and e-type by electrophysiology. Those cell types build a foundation for uncovering cellular functions, structures, and behaviors at different scales. For instance, previous correlation-based analyses found individual genes whose expression levels linearly correlate with electrophysiological features in excitatory and inhibitory neurons[3,4]. Besides, recent studies have also identified several cell types from different modalities that share many cells (e.g., me-type), suggesting the linkages across modalities in these cells[2,5]. Also, predictability from one modality to another has been found, such as predicting electrophysiological features from gene expression[6]. However, understanding the molecular mechanisms underlying multi-modalities that typically involve multiple genes is still challenging.

Transcriptomic activities such as gene expression for cellular characteristics and behaviors are fundamentally governed by gene regulatory networks (GRNs)[7]. In particular, the regulatory factors (e.g., transcription factors) in GRNs work together and control the expression of their target genes. Also, GRNs can be inferred from transcriptomic data and be employed as robust systems to infer genomic functions[8]. Many computational methods have been developed to predict the transcriptomic cell-type GRNs using single-cell genomic data such as scRNA-seq[7]. Primarily, relatively little is known about how genes function and work together in GRNs to drive cross-modal cellular characteristics (e.g., from t-type to e-type).

Further, integrating and analyzing heterogeneous, large-scale single-cell datasets remains challenging. Machine learning has emerged as a powerful tool for single-cell data analysis, such as t-SNE[9], UMAP[10], and scPred[11], to identify transcriptomic cell types. An autoencoder model has recently been used to classify cell types using multimodal data[12]. However, these studies were limited to building an accurate model as a "black box" and lacked any biological interpretability from the box, especially for linking gene expression and functional genomics to various cellular phenotypes. To address this challenge, we applied and benchmarked various machine learning methods for data alignment, including manifold learning, an emerging, and nonparametric machine learning approach, to align single-cell gene expression and electrophysiological feature data in the multiple regions of the mouse brain. We found that the nonlinear manifold alignment outperforms other methods for aligning cells from multimodalities. Also, it identified biologically meaningful cross-modal cell clusters on the latent spaces after the alignment. This finding suggests a strong nonlinear relationship (manifold structure) linking genes and electrophysiological features at the cell-type level. The aligned cells by manifold alignment show specific trajectories, suggesting the underlying gene expression transitions across neuronal cells and continuous changes of several electrophysiological features. We further found that many electrophysiological features can be predicted by differentially expressed genes of cross-modal cell clusters. Our enrichment analyses for the cell clusters, including GO terms, KEGG pathways, and gene regulatory networks, further revealed the underlying functions and mechanisms from genes to cellular electrophysiology in the mouse brain.

## Results

We have applied and benchmarked multiple existing machine learning methods to align the single cells in the mouse brain using their gene expression and electrophysiological data (Methods, Fig. 1a). In particular, we focused on two major brain regions, mouse visual cortex and motor cortex, and used the latest Patch-seq data from Allen Brain Atlas in the BRAIN Initiative[5,13,14] (Methods). The machine learning methods for alignment include linear manifold alignment (LMA) and nonlinear manifold alignment (NMA)[15], manifold warping (MW)[16], manifold alignment based on maximum mean discrepancy measure (MMD-MA)[17], unsupervised topological alignment of single-cell multi-omics integration (UnionCom)[18], Single-Cell alignment using Optimal Transport (SCOT)[19], Manifold Aligning GAN (MAGAN)[20], Canonical Correlation Analysis (CCA), Reduced Rank Regression (RRR)[5,21], Principal Component Analysis (PCA, no alignment) and t-Distributed Stochastic Neighbor Embedding (t-SNE, no alignment)[9].

The alignment methods have been previously used to align single-cell multi-omics data, e.g., scRNA-seq and scATAC-seq. Mathematically, these methods align multi-omics data of single cells and project the cells from different omics onto a latent space (e.g., co-embedding). The cells aligned on the latent space likely form certain cell clusters and share biological mechanisms, e.g., gene regulation from aligning scRNA-seq and scATAC-seq. For instance, the linear alignment methods such as canonical correlation analysis (CCA) (e.g., Seurat[22]) and RRR decompose single-cell data matrices of different omics (e.g., genes and regulatory elements across cells) to find lower-dimensional representative factors across omics. Those factors can be used to cluster cells and find the clusters' omics activities. As nonlinear alignment methods, MAGAN applies manifold alignment to match cells from single-cell multi-omics datasets using generative adversarial networks. It empirically requires biological manifolds (e.g., known cell types) to build the cell correspondences across omics for better alignment. Recently, UnionCom extends the generalized unsupervised manifold alignment (GUMA) to embed cells from each omics onto a lower-dimensional latent space (via $k$NN) and then match cross-omics spaces to align cells. Besides, Maximum Mean Discrepancy-Manifold Alignment (MMD-MA) embeds the latent spaces onto a common reproducing kernel Hilbert space by minimizing the MMD across omics. Also, SCOT uses the optimal transport technique to project one modality onto the space of another while preserving the local neighborhood of geometry from the modality. Although those methods have been shown that aligned cells have somehow specific omics activities, they have not been widely applied and tested to align additional modalities, such as gene expression vs. electrophysiology, which typically have complex and likely nonlinear cross-modal relationships (more nonlinear than cross-omics).

After benchmarking, we found that NMA better aligns cells in both regions than other methods, and also uncovers the specific trajectories of the aligned cells. Please note that NMA is nonparametric compared to other methods which are parametric. Unlike parametric methods, which are able to cross-validate learned parameters via training and testing data, nonparametric methods including NMA typically use all data samples (i.e., cells here) to directly output the cells' coordinates on the optimal aligned latent spaces.

**Manifold learning aligns single-cell multimodal data and reveals nonlinear relationships between cellular transcriptomics and electrophysiology**. For the visual cortex, after data processing and feature selection (Methods), we aligned 3654 neuronal cells (aspiny) in the mouse visual cortex using their gene expression and electrophysiological data of single cells by Patch-seq. After alignment, we projected the cells onto a low dimensional latent space and then clustered them into multiple cell

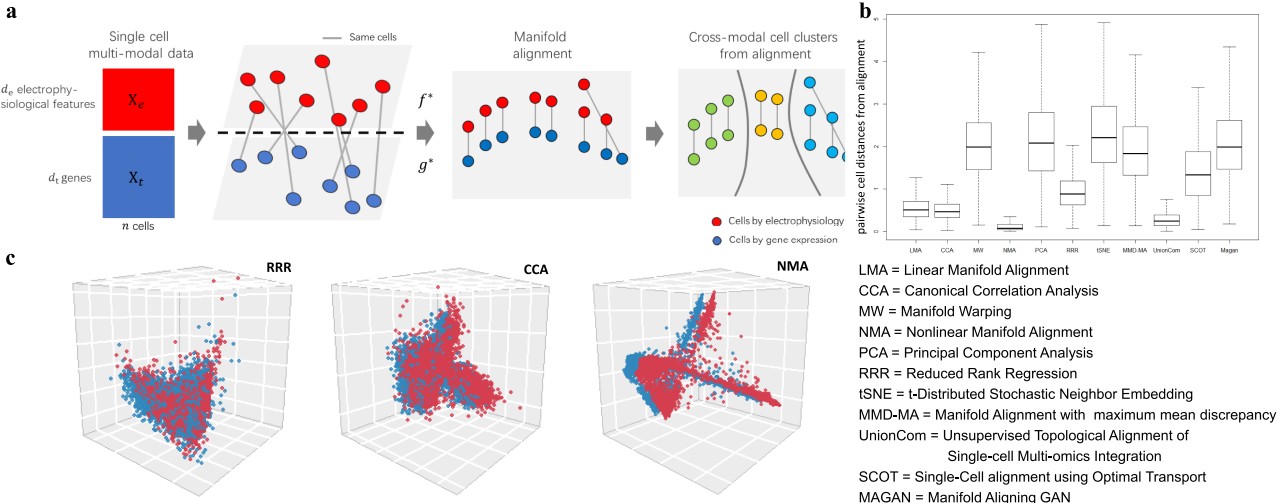

**Fig. 1 Manifold learning aligns single-cell multimodal data and reveals nonlinear relationships between cellular transcriptomics and electrophysiology.**
**a** Manifold learning analysis inputs single-cell multimodal data: $X_e$, the electrophysiological data (red,$d_e$ electrophysiological features by $n$ cells) and $X_t$, the gene expression data (blue, $d_t$ genes by $n$ cells). It then aims to find the optimal functions $f^*(.)$ and $g^*(.)$ to project $X_e$ and $X_t$ onto the same latent space with dimension $d$. Thus, it reduces the dimensions of multimodal data of n single cells to $\tilde{X}_e$ ($d$ reduced electrophysiological features by $n$ cells) and $\tilde{X}_t$ ($d$ reduced gene expression features by $n$ cells). If manifold learning is used, then the latent space aims to preserve the manifold structures among cells from each modality, i.e., manifold alignment. Finally, it clusters the cells on the latent space to identify cross-modal cell clusters. **b** Boxplots show the pairwise cell distance (Euclidean Distance) after alignment on the latent space for 3654 neuronal cells (aspiny) in the mouse visual cortex (Methods). The cell coordinates on the latent space are standardized per cell (i.e., each row of $\tilde{X} = [\tilde{X}_e, \tilde{X}_t]$) to compare methods. Each box represents one alignment method. The box indicates the lower and upper quantiles of the data, with a horizontal line at the median. The vertical line extended from the boxplot shows a 1.5 interquartile range beyond the 75th percentile or 25th percentile. The machine learning methods for alignment include linear manifold alignment (LMA), nonlinear manifold alignment (NMA), manifold warping (MW), Canonical Correlation Analysis (CCA), Reduced Rank Regression (RRR), Principal Component Analysis (PCA, no alignment), t-SNE (t-Distributed Stochastic Neighbor Embedding, no alignment), MMD-MA (Manifold Alignment with maximum mean discrepancy measurement), unsupervised topological alignment of single-cell multi-omics integration (UnionCom), Single-Cell alignment using Optimal Transport (SCOT), and Manifold Aligning GAN (MAGAN). **c** The cells on the latent space (3D) after alignment by RRR, CCA, and NMA. The red and blue dots represent the cells from gene expression and electrophysiological data, respectively. The blue dots are drifted −0.05 on the y-axis to show the alignment.

clusters. The cells clustered together imply that they share both similar gene expression and electrophysiological features. We found that nonlinear manifold alignment outperforms other methods (Fig. 1b) based on the Euclidean distances of the same cells on the latent space. We also calculated the FOSCTTM score (Methods, Fig. S1) to evaluate alignments, which also indicates that NMA performs best. Since some of the manifold alignment methods we compared are unsupervised (UnionCom, MMD-MA, SCOT), we aligned the cells in a semi-supervised fashion to compare with them, which only used 50% random cells as correlation prior and others 50% as unobserved to see how our alignment works for the cells. NMA turns out to be the second-best method, only UnionCom (average 0.280 distance and 0.060 FOSCTTM score) outperforms NMA (average 0.587 distance and 0.142 FOSCTTM score). This result suggests potential nonlinear relationships between the transcriptomics and electrophysiology in those neuronal cells, better identified by manifolds. Finally, we visualized the cell alignments of NMA, CCA, and RRR on the 3D latent space in Fig. 1c, showing that nonlinear machine learning has the best alignment (average distances of aligned same cells: RRR = 0.955, CCA = 0.510, NMA = 0.132). In addition, we applied our analysis to another multimodal data of 112 neuronal cells in the mouse visual cortex and also found that the nonlinear manifold alignment outperforms other methods (Fig. S2). Also, for the motor cortex, after aligning its 1208 neuronal cells using the gene expression and electrophysiological features (Methods), we found a similar result that the NMA outperforms other methods in terms of alignment (Fig. S3, average distances of aligned same cells: PCA = 2.366, CCA = 2.037, NMA = 0.199).

**Manifold-aligned cells recover known cell types and uncover continuous changes of electrophysiological features across transcriptomic types.** After aligning single cells using multimodal data, we found that the aligned cells on the latent space by manifold learning recovered the known cell types of a single modality. For instance, those neuronal cells were previously classified into six major transcriptomic types (t-types) or "cell classes"[14] or "cell families"[5] based on the expression of marker genes. We also found that the t-types are better formed and recovered by the latent space of NMA than other methods (e.g., CCA and RRR) (Fig. 2a, Fig. S3) in both regions. Also, since the transcriptomic types are defined by transcriptomic data, we applied PCA, t-SNE, Umap, and PHATE[39] to the transcriptomic data and UnionCom, MAGAN to both modalities of those cells and found that those methods do not show any single trajectory transitioning t-types (Fig. S4), unlike NMA (Fig. 2a). This suggests that NMA not only recovered t-types but also found a cross-t-type trajectory visualizing transitions across t-types. Using the t-types of the cells, we calculated the silhouette values of the cells on the latent space after alignment to quantify how well the coordinates of the aligned cells correspond to the t-types (Methods). We found that the silhouette values of NMA are larger than other methods (Fig. 2b), suggesting that NMA better recovers the t-types.

Also, NMA revealed an ordering across these t-types in the visual cortex (i.e., cell trajectory), implying potential gene expression transitions aligning with cellular electrophysiology. This trajectory across t-types (from Lamp5 to Vip to Serpinf1/Sncg to Sst to Pvalb) was also supported by the previous studies[23].

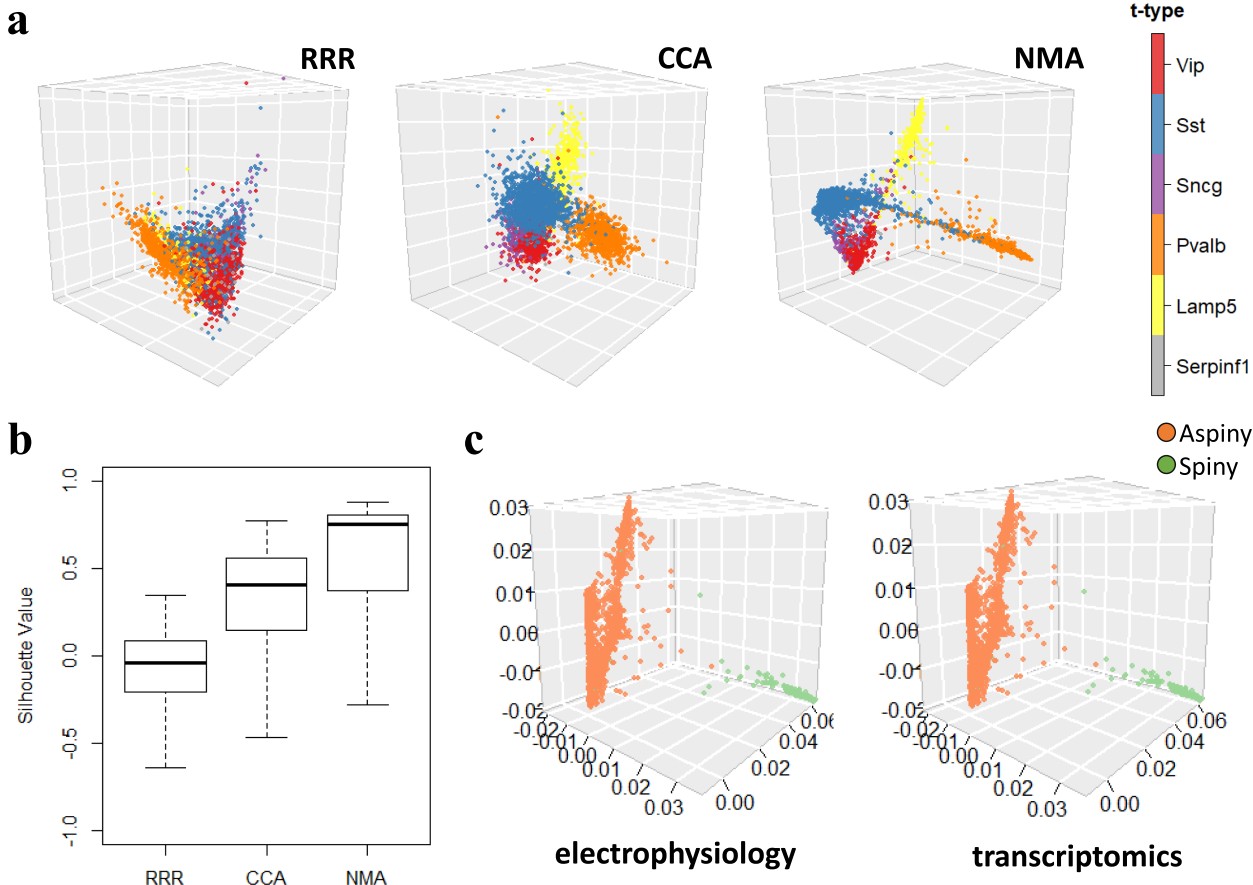

**Fig. 2 Manifold alignment of single-cell multimodalities recovers known cell types. a** Scatterplots show 3645 neuronal cells in the mouse visual cortex from electrophysiological data on the latent spaces (3D) after alignment by Reduced Rank Regression (RRR), Canonical Correlation Analysis (CCA), and Nonlinear Manifold Alignment (NMA). The cells are colored by prior known transcriptomic types (t-types). Red: Vip type; Blue: Sst type; Purple: Sncg type; Orange: Pvalb type; Yellow: Lamp5 type; Gray: Serpinf1 type. The cells from gene expression data on the latent spaces were shown in Fig. S3. **b** The boxplots show the silhouette values of cells for quantifying how well the coordinates of the cells on the latent spaces correspond to the t-types by RRR, CCA and NMA (Methods). **c** Scatterplots show neuronal cells in the mouse visual cortex on the latent spaces (3D) after alignment by NMA. Dots are colored according to the reconstructed morphological types (orange: aspiny, lightgreen: spiny).

However, other methods, including CCA, PCA, t-SNE/UMAP, and recent parametric method, reduced rank regression (RRR)[5,14], as well as recent coupled autoencoder method[12] do not show either multiple t-types or trajectories across t-types (Fig. 2a, Fig. S3). Besides t-types, the aligned cells by NMA also revealed morphological types (Methods), as shown by aspiny vs. spiny cells in Fig. 2c. Thus, these results demonstrate that manifold learning has uncovered known multimodal cell types from cell alignment. In addition, after using NMA to align cells in the motor cortex, we observed this similar trajectory (*Lamp5* to *Vip* to *Sncg* to *Sst* to *Pvalb*) (Fig. 3a).

Using NMA, we also observed trajectories in sub-t-types, implying gene expression dynamic changes and transitions across sub-t-types. For instance, the *Sst* sub-t-type is known to have multimodal diversity in Layer 5[5]. We also found that the aligned cells by NMA show a trajectory (*tac2* to *myh8* to *hpse* to *crhr2* to *chodl* to *calb2*) in both visual and motor cortices (Fig. 3b). This result suggests the great potential of NMA for revealing the underlying expression dynamics in the sub-t-types. Also, we found that certain electrophysiological features of cells on these trajectories show continuous changes. For instance, peak_t_ramp (time taken from membrane potential to AP peak for ramp stimulus) gradually changes from low to high along with the trajectory across both the t-types and *Sst* sub-t-types in the visual cortex, whereas the sag ratio changes from low to high in the

motor cortex (Fig. 3c). Also, membrane time and AP amplitude achieve high values in the middle of the trajectory across t-types in the motor cortex only (Fig. S5). These electrophysiological features' continuous changes imply the region-related activities, although both regions share similar transcriptomic trajectories.

**Cross-modal cell clusters by manifold alignment reveal genomic functions and gene regulatory networks for neuronal electrophysiology.** Furthermore, we want to systematically understand underlying functional genomics and molecular mechanisms for cellular electrophysiology using aligned cells. To this end, we clustered aligned cells on the latent space of NMA without using any prior cell-type information. In particular, we used the gaussian mixture model (GMM) to obtain five cell clusters with optimal BIC criterion (Methods, Fig. S6) in the mouse visual cortex. Those cell clusters are cross-model clusters since they are formed after aligning their gene expression and electrophysiological data. As expected, they are highly in accordance with t-types (Fig. S7). For example, Cluster 4 has ~83.3% Lamp5-type cells (373/448 cells), Cluster 2 has ~77.6% Pvalb-type cells (558/719 cells), Cluster 3 has ~86.6% Sst-type cells (1339/1546 cells) and Cluster 1 has ~79.1% Vip cells (541/684 cells). Besides, Clusters 1 and 5 include ~55.8% Serpinf1 cells (24/43) and ~60.7% Sncg cells (84/214), respectively. Moreover, we used the same clustering method to cluster the cells using a single

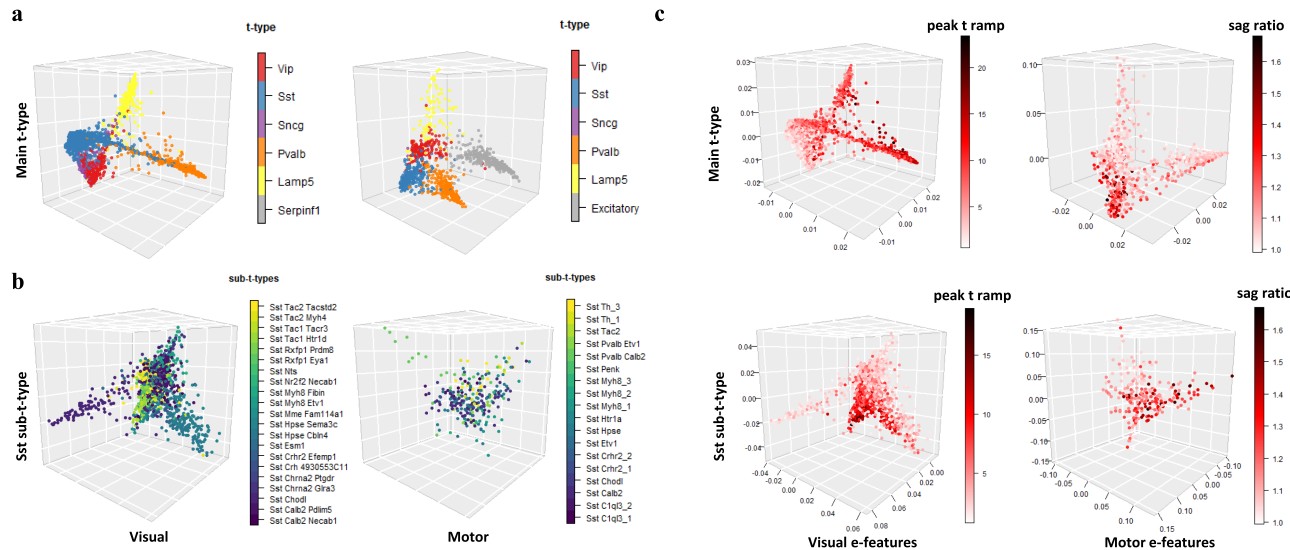

**Fig. 3 Trajectories across t-types on the latent space after nonlinear manifold alignment along with continuous electrophysiology changes. a** Scatterplots show trajectories across t-types on the latent space (3D used here) after nonlinear manifold alignment of the cells in the visual cortex (left) and motor cortex (right). Cells in shared t-types between two regions are highlighted with the same color for comparison. Red: Vip type; Blue: Sst type; Purple: Sncg type; Orange: Pvalb type; Yellow: Lamp5 type; Gray: other t-types not shared (e.g., Excitatory neurons in the visual cortex). **b** Scatterplots show trajectories for Sst sub t-types on the latent space (3D used here) after nonlinear manifold alignment of the cells in the visual cortex (left) and motor cortex (right). **c** Scatterplots show continuous changes of select electrophysiological features in t-types and Sst sub-t-types in the visual cortex (left) and motor cortex (right). The "peak t ramp" is the time taken from membrane potential to AP peak for ramp stimulus.

modality (gene expression or electrophysiology) on the PCA space without alignment. We found that those single-modal cell clusters are not so consistent with t-types as cross-modal clusters after alignment. For instance, by using electrophysiological data only, we found that the cell clusters include 57.8% Lamp5-type cells, 85.1% Pvalb-type cells, 65.1% Serpinf1-type cells, 63.1% Sncg-type cells, 49.5% Sst-type cells, and 60.8% Vip-type cells. Using gene expression data only, the cell clusters have 68.9% Lamp5-type cells, 54.4% Pvalb-type cells, 55.8% Serpinf1-type cells, 67.3% Sncg-type cells, 45.2% Sst-type cells, and 65.2% Vip-type cells. Thus, no single-modal clusters have over 70% of Vip-type, Lamp5-type and Sst-type cells. This suggests that multimodal alignment is not driven by single modalities and also helps clustering together the cells from the same types. Furthermore, in addition to GMM, we also used K-medoid and Hierarchical clustering methods to cluster aligned cells and cross-modal cell clusters. Those cross-modal clusters highly overlap with t-types as well (Fig. S8), suggesting the robustness of clustering cross-modal aligned cells. K-medoid clusters together 90.2% Lamp5-type cells, 96.6% Pvalb-type cells, 55.8% Serpinf1-type cells, 61.7% Sncg-type cells, 96.5% Sst-type cells, and 75.7% Vip-type cells. Hierarchical clustering clusters together 79.9% Lamp5-type cells, 98.6% Pvalb-type cells, 83.7% Serpinf1-type cells, 57.4% Sncg-type cells, 94.6% Sst-type cells, and 94.9% Vip-type cells.

Also, we identified differentially expressed genes (DEGs) with adjusted p-value <0.01 as marker genes of cross-modal cell clusters (Fig. 4a, Supplementary Data 1). In total, there are 182, 243, 175, 190, and 13 marker genes in Clusters 1, 2, 3, 4, 5, respectively. These cell-cluster marker genes are also enriched with biological functions and pathways (GO terms) among the genes (Supplementary Data 2) (Methods). For example, we found that many neuronal pathways and functions are significantly enriched in DEGs of Cluster 1, such as the ion channel, synaptic and postsynaptic membrane, neurotransmitter, neuroactive ligand receptor, and cell adhesion (adjusted $p < 0.05$, Fig. 4b). Further, we linked top enriched functions and pathways of each

cross-modal cell cluster to its representative electrophysiological features (Fig. 4b, Fig. S9), providing additional molecular mechanistic insights for neuronal electrophysiology. Since gene expression is fundamentally controlled by gene regulatory networks (GRNs), we predicted the GRNs for cross-modal clusters, providing mechanistic insights for multimodal characteristics (Methods). In particular, the predicted GRNs link transcription factors (TFs) to the cluster's genes (Supplementary Data 3), suggesting the gene regulatory mechanisms for the electrophysiological features in each cluster. For instance, we found that several key TFs on neuronal and intellectual development regulate the genes in Cluster 1, such as *Tcf12* and *Rora* (Fig. 4c). Also, Atf3, a TF modulating immune response[24], is regulated by inflammatory TFs, Irf5 and Spi1 in the gene regulatory networks of our clusters. Although there are cells not expressing some of these genes, due to the potential off-target expression of immunological genes in Patch-seq[25], many cells still show high and correlated expression of *Atf3*, *Irf5*, and *Spi1* (Fig. S10). This observation thus suggests potential interactions between neurotransmission and inflammation, which were recently reported[26]. Besides, *Lhx6*, a TF previously found inducing Pvalb and Sst neurons[27], was also predicted as a key TF for the Cluster 2 and Cluster 3 only that have the most Pvalb and Sst type neurons, respectively. For the motor cortex, we also identified five major cell clusters from the NMA's latent space. Like the visual cortex, the motor cortex's cell clusters also correspond to the transcriptomic types (Fig. S7). For instance, Cluster 5 has ~95.4% Vip type cells (146/153 cells). Cluster 4 has ~75.3% Sst type cells (202/271). Besides, Clusters 1 and 3 respectively include ~34.9% (101/289 cells) and ~64.7% (187/289) Pvalb type cells. For excitatory neurons, ~55.8% (218/391) of cells are in Cluster 2, and ~43.7% (171/391) of cells are in Cluster 1. The predicted GRNs for these cell clusters in the motor cortex also reveal key neuronal TFs such as *Lhx6* again, Atf4 as stress-inducible TF, and *Npdc1* for neural proliferation, differentiation, and control (Supplementary Data 3). Finally, we also predicted

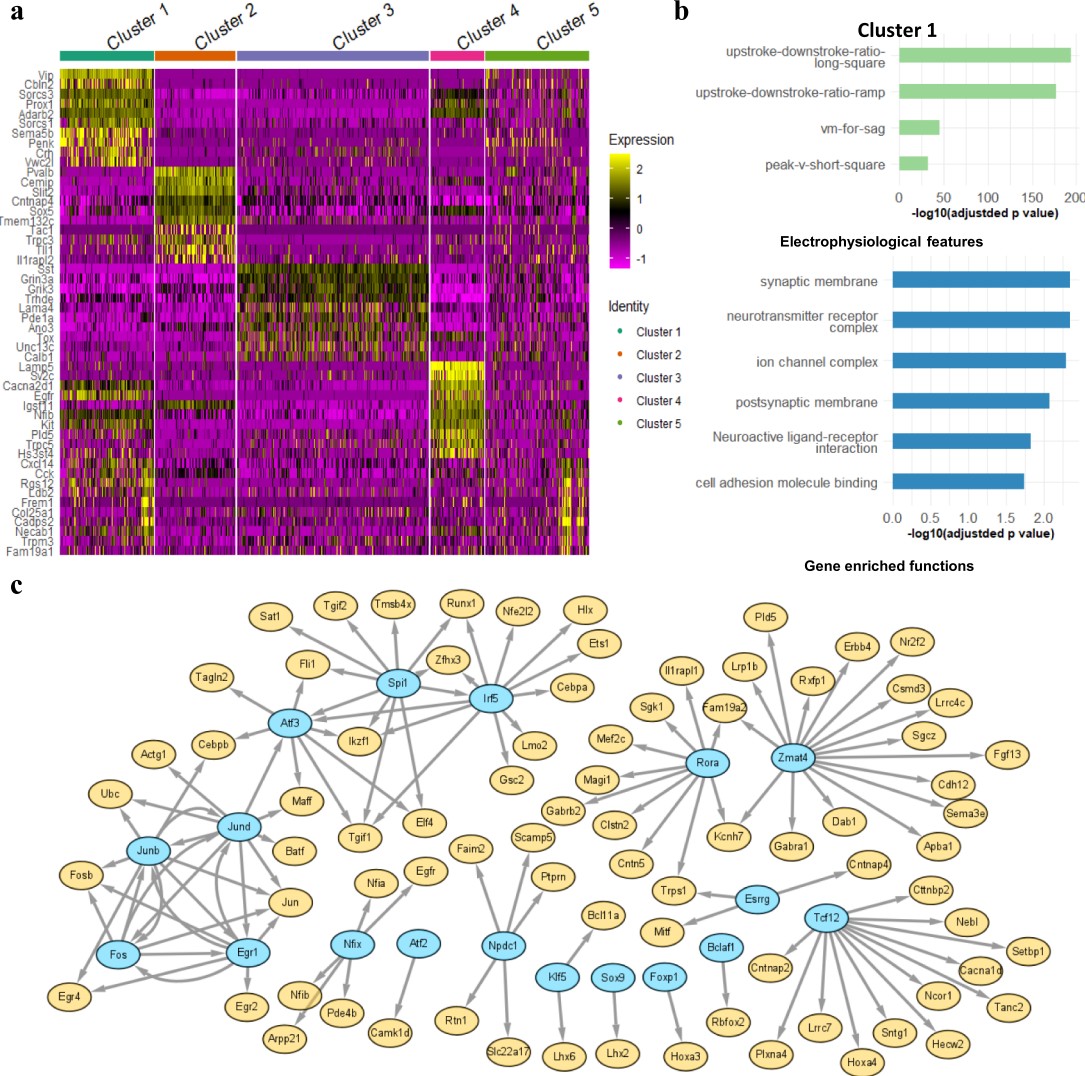

**Fig. 4 Differentially expressed genes, enrichments, and gene regulatory networks for cross-modal cell clusters. a** The gene expression levels across all 3654 cells for Top 10 differential expressed genes (DEGs) of each cross-modal cell cluster in the mouse visual cortex. The cell clusters were identified by the gaussian mixture model (Methods). The expression levels are normalized (Methods). **b** The select enriched biological functions and pathways of DEGs (GO and KEGG terms with adjusted $p$ value <0.05) and representative electrophysiological features (adjusted $p$ value <0.05) in Cluster 1 of the mouse visual cortex. **c** Gene regulatory networks that link transcription factors (TFs, cyan) to target genes (Orange) in Cluster 1 of the mouse visual cortex.

GRNs for known t-types in both regions (Supplementary Data 4), which; however, do not include several key TFs, such as *Lhx6* for Pvalb and Sst types.

**Predicting electrophysiological features from gene expression using manifold alignment results**. Finally, we want to see if the electrophysiological features could be predicted by gene expression using our manifold alignment. First, we visualized the NMA's latent spaces of the cells using the bibiplot method (a group of biplots)[5] (Fig. 5a for the visual cortex and Fig. 5b for the motor cortex). In particular, we selected the first three components of transcriptomic space and electrophysiological space so that each biplot shows such a space using two components. Due to the nonlinear manifold alignments, the transcriptomic spaces and electrophysiological spaces look much more similar than previously used linear dimensionality reduction[5]. As shown in each biplot, a group of highly correlated genes and electrophysiological features with the NMA's latent spaces are

highlighted by lines (the line length, i.e., radius, corresponds to the correlation value with max correlation = 1). We found that many genes and electrophysiological features are in similar directions in the biplots, suggesting their strong associations on the NMA's latent space. For instance, peak_t_ramp and Pavlb are in similar directions on the first and second component of the visual cortex (Fig. 5a), and peak_t_ramp indeed has high values in the Cluster 2 that is enriched with Pvalb cells (Fig. 3c, Fig. S9). Furthermore, we applied a multivariate regression model to fit the components of the NMA's electrophysiological space (dependent variables) by the components of the NMA's latent transcriptomic space (independent variables) (Methods).

Second, after showing strong associations between genes and electrophysiological features on the NMA's latent space, we next tried to predict the electrophysiological features from gene expression from our cross-modal clusters ('NMA' cell clusters, Methods) Specifically, we selected the representative electrophysiological features from each NMA cluster as potentially predictable features. We then fitted a linear regression model to

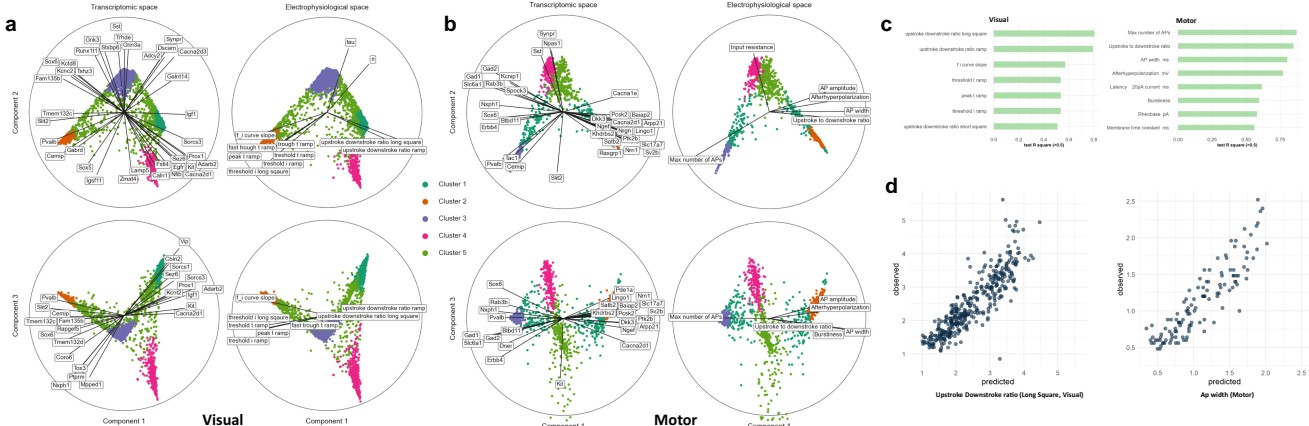

**Fig. 5 Association and prediction of electrophysiological features from gene expression. a** Bibiplots for the mouse visual cortex using the NMA's latent spaces (first three components used). Cells are dots ($n = 3654$). Transcriptomic and electrophysiological latent spaces are shown as columns. Each biplot shows the subspace of two components. Cells are colored by their cross-modal clusters. The line length of a gene or electrophysiological feature (i.e., radius) corresponds to its correlation with the latent space with max value $= 1$. The genes and electrophysiological features with correlations >0.6 are shown here. The label positions are slightly adjusted to avoid overlapping. **b** Similar to (**a**) but for the mouse motor cortex ($n = 1208$). **c** The representative electrophysiological features in the cross-modal clusters with testing $R^2 > 0.5$ (90% training set, 10% testing set, see Methods). **d** The predicted values by gene expression (x-axis) vs. the observed values (y-axis) of the upstroke downstroke ratio ($R^2 = 0.805$) in the visual cortex and the action potential width ($R^2 = 0.800$) in the mouse motor cortex.

predict each representative electrophysiological feature (dependent variable) by the expression levels of differentially expressed genes (adjusted $p$ value < 0.05) from the same NMA cluster of the feature across all cells, i.e., NMA-DEX genes. We also split the cells into 90% training and 10% testing sets and calculated the fitting $R^2$ values of testing sets (Supplementary Data 5). For example, for NMA Cluster 1 in the mouse visual cortex, we used its 182 differential expressed genes to predict the upstroke downstroke ratios for long square and ramp and achieved $R^2 = 0.805$ and 0.794, respectively. As shown in Fig. 5c, a number of electrophysiological features can be predicted by differential expressed genes of NMA cell clusters with $R^2 > 0.5$. In addition, Fig. 5d shows that the predicted values are highly correlated with the observed values across many cells for the upstroke downstroke ratio ($R^2 = 0.805$) in the visual cortex and the action potential width ($R^2 = 0.800$) in the mouse motor cortex. Moreover, we compared this result with the testing $R^2$ of predicting electrophysiological features based on the differentially expressed genes of known cell types, t-types, i.e., t-type-DEX genes. Using t-type-DEX genes, we obtained an $R^2 = 0.765$ for predicting the action potential width in the motor cortex and an $R^2 = 0.725$ for predicting the upstroke downstroke ratio for long square stimulus in the visual cortex, both of which are lower than our NMA-DEX genes. This suggests great potential of using our cross-modal clusters from nonlinear manifold alignment along with their differentially expressed genes (NMA-DEX genes) for improving predicting electrophysiological features from gene expression.

## Discussion

This study applied manifold learning to integrate and analyze single cells' gene expression and electrophysiological data in the mouse brain. We found that the cells are well aligned by the two data types and form multiple cell clusters after manifold alignment. These clusters were enriched with neuronal functions and pathways and uncovered additional cellular characteristics, such as morphology and gene expression transitions. Our manifold learning analysis is general-purpose and enables studying single-cell multimodal data in the human brain and other contexts[28].

Moreover, our GRN analysis can also serve as a basis for understanding gene regulation for additional cellular multimodal phenotypes.

Our nonlinear manifold alignment (NMA) uses the known cell correspondence information (1-to-1 from same cells) that is a unique feature of Patch-seq which simultaneously measures gene expression and electrophysiological data of the same cells. Thus, it is expected that NMA outperforms the unsupervised alignment methods, such as SCOT, MMD-MA and UnionCom. Those unsupervised methods do not need any prior knowledge on cell correspondences for alignment. Instead, they infer such correspondences in the alignment. Thus, they can be useful for aligning single-cell multimodal data when some modalities are unavailable for all cells (e.g., morphological data is only available for a fraction of cells in Patch-seq). Also, we performed a semi-supervised learning test for evaluating the alignment performance of NMA and other methods using partial cell correspondence information. We only used 1-to-1 correspondence information of 50% of 3654 neuronal cells in the mouse visual cortex to infer the correspondence of other 50% cells from alignment. As shown in Fig. S11, NMA still outperforms others except UnionCom, suggesting the potential usefulness of NMA for aligning single-cell multimodal data using partial correspondence information. Furthermore, deep-learning models have been proposed for cross-modal prediction. For example, a coupled autoencoder model[12] was proposed to align Patch-seq data to project gene expression and electrophysiological features onto two separate latent spaces. Although computationally intensive such as involving tuning many hyperparameters, given relatively large sample sizes from single-cell data, such deep-learning based models might be able to help improve multimodal data alignment in future.

Besides, this work used several electrophysiological features to represent the characteristics of neuronal electrophysiology that likely miss additional information such as continuous dynamic responses to stimulus. Thus, using advanced machine learning methods, such as deep learning for time-series classification[29] to directly integrate time-series electrophysiological data with transcriptomic data will potentially reveal deeper relationships across the modalities and improve cell-type classifications. The predicted gene regulatory networks in this study focused on linking

transcription factors to target genes on the transcriptomic side. However, gene regulation is a complex process involving many genomic and epigenomic activities, such as chromatin interactions and regulatory elements. Thus, integrating emerging single-cell sequencing data, such as scHi-C[30] and scATAC-seq[31] as additional modalities will help understand gene regulatory mechanisms in cellular characteristics and behaviors. For instance, we applied manifold learning to align co-profiled scRNA-seq and scATAC-seq data of 2,641 cells (HEK293T, NIH/3T3, A549 cells)[18]. We found that NMA still outperforms other state-of-the-arts (Fig. S12), suggesting the potential usefulness of manifold learning for additional single-cell data type integration, such as single-cell multi-omics data and understanding single-cell functional genomics.

## Methods
**Single-cell multimodal datasets**. We applied our machine learning analysis for multiple single-cell multimodal datasets in the mouse brain.

*Visual cortex.* Primarily, we used a Patch-seq dataset that included the transcriptomic and electrophysiological data of 4435 neuronal cells (GABAergic cortical neurons) in the mouse visual cortex[14]. In particular, the electrophysiological data measured multiple hyperpolarizing and depolarizing current injection stimuli and responses of short (3 ms) current pulses, long (1 s) current steps, and slow (25 pA/s) current ramps. The transcriptomic data measured genome-wide gene expression levels of those neuronal cells. Six transcriptomic cell types (t-types) were identified among the cells: Vip, Sst, Sncg, Serpinf1, Pvalb, and Lamp5. Further, morphological information was provided: 4293 aspiny and 142 spiny cells. Also, we tested our analysis for another Patch-seq dataset in the mouse visual cortex[13]. This dataset includes 112 neuronal cells with electrophysiological data and gene expression data (Fig. S2).

*Motor cortex.* Another Patch-Seq dataset included the transcriptomic and electrophysiological data of 1227 neuronal cells (GABAergic cortical neurons) in the mouse motor cortex[5]. The electrophysiological data measured multiple hyperpolarizing and depolarizing current injection stimuli and responses of long current steps. The transcriptomic data measured genome-wide gene expression levels of those neuronal cells. Five major transcriptomic cell types (t-types) were identified among the cells: Vip, Sst, Sncg, Pvalb, and Lamp5, based on which 90 neuronal sub-t-types were also labeled.

### Data processing and feature selection of multimodal data
*Visual cortex.* For electrophysiology, we first obtained 47 electrophysiological features (e-features) on stimuli and responses, which were identified by Allen Software Development Kit (Allen SDK) and IPFX Python package[32]. Second, we eliminated the features with many missing values such as short_through_t and short_-through_v, as well as the cells with unobserved features, and finally selected 41 features in all three types of stimuli and responses for 3654 aspiny cells (inhibitory) and 118 spiny cells (excitatory) out of the 4435 neuronal cells. Since the spiny cells usually do not contain the t-type information, we used the 3654 aspiny cells for manifold learning analysis. Together, we used the 3654 aspiny cells and 118 spiny cells to refer to morphological cell types (m-type). Also, we standardized the feature values across all cells to remove potential scaling effects across features for each feature. The final electrophysiological data matrix is $X_e$ (3654 cells by 41 e-features). We selected 1302 neuronal marker genes[33] and then took the log transformation of their expression levels. The final gene expression data is $X_t$ (3654 cells by 1302 genes).

*Motor cortex.* For electrophysiology, there are 29 electrophysiological features summarized by[5]. We eliminated the cells with missing observations in these features and standardized them across each feature. Then we selected 1208 cells with features aroused by long square stimuli. For gene expression data, we again selected 1329 neuronal marker genes[33] and then took the log transformation of their expression levels. The final electrophysiological data matrix is $X_e$ (1208 cells by 29 e-features), and the gene expression data is $X_t$ (1208 cells by 1329 genes).

### Manifold learning for aligning single cells using multimodal data. We applied
our published tool, ManiNetCluster[34] to perform various manifold learning approaches to align single cells using their multimodal data to discover the linkages of genes and electrophysiological features, including linear manifold alignment (LMA) and nonlinear manifold alignment (NMA)[15], manifold warping (MW)[16]. In particular, the manifold alignment projects the cells from different modalities onto a lower-dimensional common latent space for preserving the local similarity of cells in each modality (i.e., manifolds). The distances of the same cells on the latent space can quantify the performance of the alignment. Mathematically, given $n$

single cells, let $X_e = [x_e^1, \dots, x_e^n] \in \mathbb{R}^{d_1 \times n}$ and $X_t = [x_t^1, \dots, x_t^n] \in \mathbb{R}^{d_2 \times n}$ represent their electrophysiological and gene expression matrices, respectively, where $d_1$ is the number of electrophysiological features, and $d_2$ is the number of genes. The manifold alignment finds the optimal projection functions $f^*(.)$ and $g^*(.)$ to map $x_e^i, x_t^i$ onto a common latent space via manifolds with dimension $d << min(d_1, d_2)$:

$$f^*, g^* = \arg\min_{f,g}(1-\mu)\sum_{i=1}^n\sum_{j=1}^n ||f(x_e^i) - g(x_t^j)||_2^2 W^{i,j}$$
$$+ \mu\sum_{i=1}^n\sum_{j=1}^n ||f(x_e^i) - f(x_e^j)||_2^2 W_{X_e}^{i,j} \qquad (1)$$
$$+ \mu\sum_{i=1}^n\sum_{j=1}^n ||g(x_t^i) - g(x_t^j)||_2^2 W_{X_t}^{i,j}$$

where the corresponding matrix $W \in \mathbb{R}^{n \times n}$ models cross-modal relationships of cells (i.e., identity matrix here), and the similarity matrices $W_{X_e}, W_{X_t} \in \mathbb{R}^{n \times n}$ model the relationships of the cells in each modality and can be identified by $k$-nearest neighbor graph ($k$NN, matrix elements between neighbors =1 and otherwise = 0). As shown on Fig. S13, we tried different values of $k$ ($k$ = 2, 5, 8, 10) and $d$ ($d$ = 3, 5, 8, 10) and found that as $k$ and $d$ grow, the distances of aligned cells did not change much and NMA always outperforms others. Thus, we used $k$=2 and $d$ = 3, which achieve the minimum average distance among the same cells. The parameter $\mu$ trades off the contribution between the preserving local similarity for each modality (intra-modal) and the correspondence of the cross-modal network (inter-modal). We used $\mu = 0.5$ to balance two losses. Moreover, this also makes our alignment comparable with other methods, such as MMD-MA, UnionCom, SCOT, and MAGAN, all of which also assign equal weights to all losses from intra- and inter-modal contributions.

In addition, to avoid finding all-zero solutions, we have to add the non-zero constraint while solving this minimization: $Q^T D Q = I$, where $Q = \begin{bmatrix} f \\ g \end{bmatrix}$,

$f = [[f_k(x_e^1)\dots f_k(x_e^{d_1})]]_{k=1}^d$, $g = [[g_k(x_t^1)\dots g_k(x_t^{d_1})]]_{k=1}^d$, $D$ is the diagonal matrix of $\mu W_{X_e}, \mu W_{X_t}$, and $I$ is the identity matrix. Again, we used our previous ManiNetCluster method[31] to solve this optimization and found the optimal functions and latent spaces for aligned cells using linear and nonlinear methods, including linear manifold alignment, canonical correlation analysis, linear manifold warping, nonlinear manifold alignment, and nonlinear manifold warping. Finally, after alignment, let $\tilde{x}_e^i = f^*(x_e^i) \in \mathbb{R}^d$ and $\tilde{x}_t^i = g^*(x_t^i) \in \mathbb{R}^d$ represent the coordinates of the $i_{th}$ cell on the common latent space ($d$-dimension) and $d$ be 3 in our analysis for visualization. Moreover, the nonlinear manifold alignment is nonparametric and directly outputs the coordinates of the cells on the optimal latent spaces, without explicitly providing optimal mapping functions. In addition to the pairwise distances of cells on the common latent space, we also used the metric, fractions of samples closer than the true match (FOSCTTM)[18] for evaluation. In particular, we calculated the FOSCTTM score of aligned cells as follows. For each cell in the electrophysiological data, we first find its true match in the gene expression data, then rank all other cells on the aligned latent space based on their distances from $x$, and finally compute the fraction of cells that are closer than the true match.

### Identification of cross-modal cell clusters using Gaussian Mixture Model.
After NMA alignment, the cells clustered together on the latent space imply that they share similar transcriptomic and electrophysiological features and form cross-modal cell clusters ('NMA' cell clusters). To identify such cross-modal cell clusters, we clustered the cells on the latent space into the cell clusters using gaussian mixture models (GMM) with $K$ mixture components. Given a cell, we assigned it to the component $k_0$ with the maximum posterior probability:

$$\Pr(k_0|\tilde{X}_{et}^i, \lambda) = \frac{w_i g(\tilde{X}_{et}^i|\sum_{k_0}, k_0)}{\sum_{k=1}^K w_k g(\tilde{X}_{et}^i|\sum_k, k)} \qquad (2)$$

where $\tilde{x}_{et}^i$ is the $i^{th}$ row of a combined feature set $[\tilde{X}_e, \tilde{X}_t]$, $\lambda = \{w_k, \mu_k|\Sigma_k\}k = 1, \dots, K$ are parameters: mixture weights, mean vectors, and covariance matrices. Finally, the cells assigned to the same component form a cross-modal cell type. Also, we used the Expectation-maximization algorithm (EM) algorithm with 100 iterations to determine the optimal number of clusters with $K = 5$ (Fig. S6) by Bayesian information criterion (BIC) criterion[35]. $K = 5$ was chosen at which the $BIC = Kln(n) + 2(\hat{L})$ of the model has an approximately constant and insignificant gradient descent through the equation. Silhouette values are used to compare the clustering result[36], which takes a value from $-1$ to 1 for each cell and indicates a more pronouncedly clustered cell as the value increases.

### Differentially expressed genes, enrichment analyses, gene regulatory networks, and representative cellular features of cross-modal cell clusters. We
used the Seurat to identify differentially expressed genes of each cell cluster and multiple tests, including Wilcox and ROC, to further identify the marker genes of cell clusters (adjusted $p$ value < 0.01)[22]. We applied this method to the electrophysiological features (absolute values) to find each cluster's represented e-features. Also, we used the web app, g:Profiler to find the enriched KEGG pathways, GO

terms of cell-cluster marker genes, implying underlying biological functions in the cell clusters[37]. Enrichment $p$ values were adjusted using the Benjamin–Hochberg (B–H) correction. Furthermore, we predicted the gene regulatory networks for cell clusters, linking transcription factors to target marker genes by SCENIC[38]. Those networks provide potentially additional regulatory mechanistic insights for electrophysiology at the cell-type level.

**Prediction of electrophysiological features using gene expression**. We generated the bibiplots that consist of a group of biplots using the method in[5]. In particular, we used the first three components of the transcriptomic and electrophysiological latent spaces from nonlinear manifold alignment (NMA) as the latent spaces for generating biplots. For the multivariate linear regression, let $\tilde{X}_e \in \mathbb{R}^{n \times d}$ and $\tilde{X}_t \in \mathbb{R}^{n \times d}$ be the first $d$ dimensions of the electrophysiological and transcriptomic latent spaces, respectively, where $n$ is the number of cells for training. The loss function of the multivariate regression is defined as $\mathcal{L} = \|\tilde{X}_e - \tilde{X}_t B\|^2$, and the solution is given by $\hat{B} = (\tilde{X}_t^\top \tilde{X}_t)^{-1} \tilde{X}_t^\top \tilde{X}_e$. Also, we performed 10-fold cross-validation with 20 repetitions. For each repetition, all cells were randomly partitioned into 10 subsets. A subset was selected as a testing set, and the remaining subsets were assigned as training sets. The training sets were used to estimate coefficients, and the testing set was used to calculate $R^2$. The process was repeated 10 times to choose different testing sets. Cross-validated $R^2$ is calculated through $R^2 = 1 - \frac{\|\tilde{X}_e^{test} - \tilde{X}_t^{test}\hat{B}\|^2}{\|\tilde{X}_e^{test}\|^2}$, where $\tilde{X}_e^{test}$ and $\tilde{X}_t^{test}$ were centered using testing set means. The reported $R^2$ is averaged across all folds and repetitions. We also tried multiple $d$ values to check where the regression overfits, especially for the low dimensionality of the latent space. We varied $d$ from 3 to 20 and found that the cross-validated $R^2$ does not change too much and slightly decreased as the dimension increases (from 0.987 to 0.954 for the visual cortex; from 0.977 to 0.952 for the motor cortex).

Also, we used the multivariate linear regression to predict represented electrophysiological features by the expression levels of differentially expressed genes (DEGs) (adjusted $p$ value < 0.05) of our cross-modal clusters (and t-types). In particular, we split the cells into 90% training set ($n_{train}$ cells) and 10% testing set ($n_{test}$ cells). Let $X_{t_i} \in \mathbb{R}^{n_{train} \times c_i}$ represent the expression levels of $c_i$ differential expressed genes in Cluster $i$, and $Y_{ij} \in \mathbb{R}^{n_{train}}$ represent the observed values of $j$th represented electrophysiological feature in Cluster $i$ for the training cells, the predicted $j$th electrophysiological feature $\hat{Y}_{ij} \in \mathbb{R}^{n_{train}}$ and regression parameters $\hat{\beta}_{ij}$ given by $\hat{Y}_{ij} = X_{t_i}\hat{\beta}_{ij} = X_{t_i}(X_{t_i}^\top X_{t_i})^{-1} X_{t_i}^\top Y_{ij}$, based on the solution to the multivariate linear regression as above. Finally, we can predict the electrophysiological feature for testing set, $\hat{Y}_{ij}^{test} \in \mathbb{R}^{n_{test}}$ by $\hat{Y}_{ij}^{test} = X_{t_i}^{test}\hat{\beta}_{ij}$, and calculate both the training and testing $R^2$ values for evaluating the prediction.

**Statistics and reproducibility**. Differentially gene expression analysis was implemented by Seurat[22] (adjusted $p$ value < 0.01). Gene set enrichment analysis was done by the web app, g:Profiler[37]. Enrichment p-values were adjusted using the Benjamin–Hochberg (B–H) correction. Silhouette values were calculated by R function silhouette(). Gaussian mixture models for clustering were implemented by R package gmm.

**Reporting Summary**. Further information on research design is available in the Nature Research Reporting Summary linked to this article.

## Data availability

All our results are provided in Supplementary Data 1–5. All processed data are available at https://github.com/daifengwanglab/scMNC. All other data are available from the corresponding author on reasonable request.

## Code availability

The codes for our analyses and figures are available at https://github.com/daifengwanglab/scMNC.

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

## Acknowledgements
This work was supported by National Institutes of Health grants, R01AG067025, R21CA237955, R03NS123969 and U01MH116492 to D.W., P50HD105353 to Waisman Center, and the start-up funding for D.W. from the Office of the Vice Chancellor for Research and Graduate Education at the University of Wisconsin–Madison.

## Author contributions
D.W. conceived and designed the study. J.H., J.S. and D.W. analyzed the data and wrote the manuscript. All authors read and approved the final manuscript.

## Competing interests
The authors declare no competing interests.
