## [Transparent Peer Review File · Communications Biology]

Reviewers' comments:

Reviewer #1 (Remarks to the Author):

The authors focused on manifold learning for integrating single-cell multi-modal data, which is a computationally urgent task and should be of general interest to the community. They applied and benchmarked multiple machine learning methods to align gene expression and electrophysiological data of single neuronal cells in mouse brain, and showed that the nonlinear manifold learning method from their published tool outperformed other methods. Further downstream analysis was performed based on the nonlinear manifold alignment results.

Overall, the authors addressed an important topic, and this manuscript is well-written and clear. However, I have some major concerns, as discussed below.

One concern regarding results is that current study is restricted to aligning single-cell Patch-seq datasets of neuronal electrophysiology and transcriptomics. However, when applying these methods to other types of single-cell multi-modal data (such as scRNAseq and scATACseq datasets), it would be interesting to see whether the conclusion in current manuscript still applies. This will help the users to be able to fully assess the usefulness of manifold learning methods for single-cell data integration.

Moreover, the manuscript lacks a detailed comparison between integration methods, outlining the specific aspects in which their proposed nonlinear manifold alignment method might perform better or worse than other methods (e.g., MMD-MA, a kernel based nonlinear manifold method). This makes it slightly harder for users to obtain a general conclusion from the study.

Minor issues:

Some recent single-cell manifold alignment references, such as SCOT and UnionCom, are not cited. It would be more interesting to see the performance benchmarks with these methods.

The authors may need to clearly present how the hyperparameters of the applied methods were determined given datasets.

Reviewer #2 (Remarks to the Author):

Summary:

This paper applies manifold alignment algorithms to learn a cross-modal embedding combining gene expression and electrophysiological data of single neuronal cells in the mouse brain from Brain Initiative. The dataset is obtained from a method called Patch-seq that simultaneously measures the transcriptome and electrophysiological features of a single cell. Therefore, this dataset has 1-1 correspondence between the cells. The alignment methods explored in this work include linear manifold alignment (LM), nonlinear manifold alignment (NMA), manifold warping (MW), and manifold alignment based on maximum mean discrepancy (MMD-MA). Other methods like Canonical Correlation Analysis, Reduced Rank Regression (RRR), Principal Component Analysis (PCA, without alignment), and t-Distributed Stochastic Neighbor Embedding (t-SNE, without alignment) are also used. The paper reports best alignment results for the NMA method using the euclidean distance metric between true cell pairs across gene expression and electrophysiological data modalities. NMA is also able to preserve the known cell types in the aligned space, which is measured by using silhouette scores. Finally, the paper presents results of downstream analysis of the aligned datasets to demonstrate the biological significance of performing such alignments. Overall, the paper makes a reasonable case for performing alignments across modalities in single-cell datasets. However, it requires robust evaluation metrics and additional experiments to strongly support its claim regarding the accuracy of the alignment algorithms and the usefulness of the information captured by the alignment. Please see the

detailed comments below.

Strengths:

++ The paper presents a useful application of the manifold alignment algorithms for analyzing the output of the Patch-seq datasets obtained from the mouse brain. Previous methods working with this dataset have focused on the cross-modal prediction task (Gala et al., 2021), while this paper presents results for cross-modal alignment of the transcriptome and electrophysiological datasets of single-cell.

++ The paper includes 8 different methods to show that overall NMA provides the best alignment performance and then proceeds to use the results from it for downstream analysis.

++ The paper uses silhouette score metric to show that the cell types are preserved in the NMA aligned space. Furthermore, one can find trajectories in the transcriptome data and by mapping the electrophysiological data modality to it, can uncover the same for it.

++ The unsupervised clustering on the aligned space reveals biological mechanisms through GO and GRN analysis

Weaknesses:

-- The NMA method uses the 1-1 correspondence information when aligning the datasets and gets better performance over MMD-MA, which is an unsupervised algorithm that does not use any correspondence information for alignment. Therefore, the comparison might be a bit misleading. Furthermore, for all the methods applied in the paper, was a reasonable number of hyperparameters explored before picking the final best-performing results? If so, these details should be presented.

-- On a similar note, the paper chose a lower dimension size of $d=3$ to align the modalities, however, one may argue that the best alignment for some of the other methods may be in a different lower-dimensional space (say $d=5$). One can use PCA to project the best alignment in $d=3$ space for visualization but restricting the alignment space to $d=3$ may not be ideal for all the methods.

-- The paper explores the k hyperparameter of the NMA algorithm. It is unclear if something similar was done for μ and d .

-- The paper does not mention MAGAN (Amodio et al, 2018) and UnionCom (Cao et al., 2020) which are also manifold alignment methods, with the former using a deep learning architecture and the latter a variation of Generalized Unsupervised Manifold Alignment (Cui et al. 2014)

-- Is there a specific reason why the focus is on manifold alignment methods since multiple methods have been proposed for alignment of data modalities with 1-1 supervision like Seurat V4 (Hao et al.), LIGER (Welch et al. 2019), etc.? If so, the paper must elaborate on that.

-- The evaluation metric for evaluating the alignment performance is euclidian distance. Technically, the true pair might be distant from each other in the aligned space, but the alignment still works if a sample point is closer to its true match compared to other sample points. Therefore, a metric like fractions of samples closer than the true match (FOSCTTM), used for performance evaluation of MMD-MA and UnionCom might be better to evaluate the performance. It might also be worth reporting the Label Transfer Accuracy metric (used in UnionCom paper).

-- The significance of the downstream analysis of NMA space is hard to gauge. It would help to show that if one was to perform a similar analysis on the individual PCA of the two modalities (without alignment), they will not be as informative. Or one modality may turn out to be informative (like transcriptome) but the other will require alignment to be as informative. It is most likely the case but

there are no experiments to highlight this. One could also do this to compare other alignment methods.

-- It would also be useful to discuss how do the cross-modality prediction results compare to the one reported in Gala et al. (2021)

Minor comments:

Brief descriptions of some of the other manifold alignment methods applied would be very useful.

It is likely that I missed something in the description and would like to clarify my confusion regarding the cross-modal prediction result. Given that the NMA method explicitly matches the features across the two datasets, it is unclear what is the significance of reporting high correlation when regression is performed using the components of the NMA space? Is it not to be expected?

Reviewer #3 (Remarks to the Author):

This paper performs a benchmarking study of various single-cell alignment methods. I have multiple major concerns with this study.

The biggest concern is with the comparison measures. The NMA method is purely supervised in the sense the cross-correspondence matrix is given a priori. Thus it is expected for the same cell distances in the final embedding to be small. Thus, the boxplot fig 1B is not relevant. You can achieve a perfect score (almost 0 distance between the corresponding cells, by just setting the u parameter close to 0). For instance, MMD-MA is unsupervised, therefore it should work worse for this kind of comparison against a supervised method. Thus the comparisons between the supervised and unsupervised methods are unfair.

An alternative approach would be to perform the algorithm using just a small set of cross-correspondence relationships, in a more realistic semi-supervised fashion and compare it to (Cao, K., 2020, reference below). You could do this measuring the Euclidean distances of the same cells on the latent space, but for the observations not considered in the construction of the cross-correspondence matrix.

There are also some remaining questions about the approach. Does the construction of the similarities matrices for each modality use a kernel? Or is it just a 1-0 matrix indicating the nearest neighbors?. More explanation about this is required, especially since what might happen in the objective function is that the similarities of some modality might be much stronger than in the other. And in such a scenario, the objective function might just prefer to take one of the modalities into account, and in junction with the cross-correspondence similarities (which might be the strongest part of the objective function), the method would just create an embedding of the transcriptomics (or the electrophysiology) and then "match" the cells, not taking into account the electrophysiology features (or the transcriptomics). In this sense, the separate embeddings of each domain before alignment are needed to assess the impact of the alignment (see some examples of this in the suggested references below).

Regarding this previous comment, the clustering comparison might be biased. The final embedding might prefer the transcriptomics domain keeping a good separation between t-types. Given that the paper aims to do benchmarking the comparison should be more extensive. By using different clustering approaches and performance measures, this would give a more general idea of the robustness of different methods.

Some other questions/comments:

When the authors claimed t-types trajectories are not recovered by other methods, did you tried PHATE in the transcriptomic domain? PHATE often does well at visualizing trajectory data.

Is there a particular reason to use the Laplacian eigenmaps objective function for each modality? It could be interesting to use PHATE or tsne/umap, since those often work better.

If the goal is to benchmark the methods more discussion of them is required in similar detail as the authors did for NMA.

The authors should also compare to the following methods:

Liu, J., Huang, Y., Singh, R., Vert, J. P., & Noble, W. S. (2019). Jointly embedding multiple single-cell omics measurements. *BioRxiv*, 644310.

Cao, K., Bai, X., Hong, Y., & Wan, L. (2020). Unsupervised topological alignment for single-cell multi-omics integration. *Bioinformatics*

Amodio, M., & Krishnaswamy, S. (2018, July). MAGAN: Aligning biological manifolds. In *International Conference on Machine Learning*

Cui, Z., Chang, H., Shan, S., & Chen, X. (2014). Generalized unsupervised manifold alignment. *Advances in Neural Information Processing Systems*, 27, 2429-2437.

We have addressed the reviewer comments and provided the point-by-point responses as follows. We really appreciate the work of reviewers to help improve our manuscript.

Reviewer 1

-- Ref 1.0 Summary --

Reviewer Comment	The authors focused on manifold learning for integrating single-cell multi-modal data, which is a computationally urgent task and should be of general interest to the community. They applied and benchmarked multiple machine learning methods to align gene expression and electrophysiological data of single neuronal cells in mouse brain, and showed that the nonlinear manifold learning method from their published tool outperformed other methods. Further downstream analysis was performed based on the nonlinear manifold alignment results. Overall, the authors addressed an important topic, and this manuscript is well-written and clear. However, I have some major concerns, as discussed below.
Author Response	Thank the reviewer for the great summary. We appreciate that the reviewer acknowledges that we addressed an important topic, and the manuscript was well-written and clear. We have addressed the concerns as follows.

-- Ref 1.1 Manifold alignment of scRNA-seq and scATAC-seq data --

Reviewer Comment	One concern regarding results is that current study is restricted to aligning single-cell Patch-seq datasets of neuronal electrophysiology and transcriptomics. However, when applying these methods to other types of single-cell multi-modal data (such as scRNAseq and scATACseq datasets), it would be interesting to see whether the conclusion in the current manuscript still applies. This will help the users to be able to fully assess the usefulness of manifold learning methods for single-cell data integration.
Author Response	We thank the reviewer for suggesting applying manifold learning to align scRNA-seq and scATAC-seq data. Although this paper focuses on analyzing neuronal transcriptomic and electrophysiological data, as requested, we applied manifold alignment to a dataset co-profiling scRNA-seq and scATAC-seq of 2,641 cells (HEK293T, NIH/3T3, A549 cells) (Cao et al., Science, 2018 and GEO: GSM3271040, GSM3271041). We found that nonlinear manifold alignment (NMA) still outperforms other state-of-the-arts for aligning single cells (new Fig.

S9), suggesting potential usefulness of manifold learning for additional single-cell data type integration. We also included this analysis into the manuscript as below.

Excerpt
From
Revised
Manuscript

Line 335, Page 9 “For instance, we applied manifold learning to align co-profiled scRNA-seq and scATAC-seq data of 2,641 cells (HEK293T, NIH/3T3, A549 cells) 18. We found that NMA still outperforms other state-of-the-arts (Fig. S12), suggesting potential usefulness of manifold learning for additional single-cell data type integration such as single-cell multi-omics data and understanding single-cell functional genomics.”

Figure S12 Pairwise Euclidean distance of aligned scATAC-seq and scRNA-seq data of 2,641 cells (HEK293T, NIH/3T3, A549 cells). The co-profiling data of scRNA-seq and scATAC-seq was generated by the assay, sci-CAR (Cao et al., Science, 2018) and downloaded from GEO: GSM3271040 and GSM3271041. The machine learning methods for alignment include linear manifold alignment (LMA), Canonical Correlation Analysis (CCA), manifold warping (MW), nonlinear manifold alignment (NMA), UnionCom, SCOT, Magan, MMD-MA, Principal Component Analysis (PCA, no alignment), and reduced rank regression (RRR), and t-Distributed Stochastic Neighbor Embedding (t-SNE, no alignment).

-- Ref 1.2 Comparing other integration methods --

Reviewer
Comment

Moreover, the manuscript lacks a detailed comparison between integration methods, outlining the specific aspects in which their proposed nonlinear manifold alignment method might perform better or worse than other methods (e.g., MMD-MA, a kernel based nonlinear manifold method). This makes it slightly harder for users to obtain a general conclusion from the

	study.
Author Response	We thank the reviewer for pointing out this concern. To address this concern, we also benchmarked many recent methods (e.g., MMD-MA, UnionCom, SCOT, MAGAN, see Ref 1.3) and found that NMA still outperforms them for aligning Patch-seq data of the mouse visual cortex. Also, we provided a detailed introduction about these methods in the main text.
Excerpt From Revised Manuscript	Line 76, Page 2: “We have applied and benchmarked multiple existing machine learning methods to align the single cells in the mouse brain using their gene expression and electrophysiological data (Methods, Fig. 1A). In particular, we focused on two major brain regions, mouse visual cortex and motor cortex, and used the latest Patch-seq data from Allen Brain Atlas in the BRAIN Initiative ^{5,13,14} (Methods). The machine learning methods for alignment include linear manifold alignment (LMA) and nonlinear manifold alignment (NMA) ¹⁵, manifold warping (MW) ¹⁶, manifold alignment based on maximum mean discrepancy measure (MMD-MA) ¹⁷, unsupervised topological alignment of single-cell multi-omics integration (UnionCom) ¹⁸, Single-Cell alignment using Optimal Transport (SCOT) ¹⁹, Manifold Aligning GAN (MAGAN) ²⁰, Canonical Correlation Analysis (CCA), Reduced Rank Regression (RRR) ^{5,21}, Principal Component Analysis (PCA, no alignment) and t-Distributed Stochastic Neighbor Embedding (t-SNE, no alignment) ⁹. The alignment methods have been previously used to align single-cell multi-omics data, e.g., scRNA-seq and scATAC-seq. Mathematically, these methods align multi-omics data of single cells and project the cells from different omics onto a latent space (e.g., co-embedding). The cells aligned on the latent space likely form certain cell clusters and share biological mechanisms, e.g., gene regulation from aligning scRNA-seq and scATAC-seq. For instance, the linear alignment methods such as canonical correlation analysis (CCA) (e.g., Seurat ²²) and RRR decompose single-cell data matrices of different omics (e.g., genes and regulatory elements across cells) to find lower-dimensional representative factors across omics. Those factors can be used to cluster cells and find the clusters’ omics activities. As nonlinear alignment methods, MAGAN applies manifold alignment to match cells from single-cell multi-omics datasets using generative adversarial networks. It empirically requires biological manifolds (e.g., known cell types) to build the cell correspondences across omics for better alignment. Recently, UnionCom extends the generalized unsupervised manifold alignment (GUMA) to embed cells from each omics onto a lower-dimensional latent space (via kNN) and then match cross-omics spaces to align cells. Besides, Maximum Mean Discrepancy-Manifold Alignment (MMD-MA) embeds the latent spaces onto a common reproducing kernel Hilbert space by minimizing the MMD across omics. Also, SCOT uses the optimal transport technique to project one modality onto the space of another while preserving the local neighborhood of geometry from the modality. Although those methods have been shown that aligned cells have

somehow specific omics activities, they have not been widely applied and tested to align additional modalities such as gene expression vs. electrophysiology, which typically have complex and likely nonlinear cross-modal relationships (more nonlinear than cross-omics).”

-- Ref 1.3 Benchmarking more recent methods --

Reviewer Comment	Some recent single-cell manifold alignment references, such as SCOT and UnionCom, are not cited. It would be more interesting to see the performance benchmarks with these methods.
Author Response	Thank the reviewer for suggesting recent alignment methods. As suggested, we benchmarked additional methods, SCOT, UnionCom and MAGAN for the mouse visual cortex data. We found that NMA still outperforms all others in terms of both pairwise distances (updated Fig. 1B) and fractions of samples closer than the true match (FOSCTTM, suggested by Ref 2.6, new Fig. S1). We also introduced those methods in the Introduction of the main text.
Excerpt From Revised Manuscript	 The figure is a box plot comparing the performance of eleven different manifold alignment methods. The y-axis represents the 'pairwise cell distances from alignment', ranging from 0 to 5. The x-axis lists the methods: LMA, CCA, MW, NMA, PCA, RRR, tSNE, MMD-MA, UnionCom, SCOT, and Magan. Each method is represented by a box plot showing the median (horizontal line inside the box), the interquartile range (the box itself), and the full range of the data (whiskers). NMA consistently shows the lowest median distance and the smallest interquartile range, indicating it provides the most accurate pairwise alignments among the methods tested. Other methods like PCA, tSNE, and Magan show significantly higher median distances and larger spreads, suggesting poorer alignment performance.

Figure 1B (top) and **Figure S1** (bottom) show the pairwise cell distance (Euclidean Distance) and fractions of samples closer than the true match (FOSCTTM), respectively, after alignment on the latent space for 3654 neuronal cells (aspiny) in the mouse visual cortex. The cell coordinates on the latent space are standardized per cell (i.e., each row of $\tilde{X} = [\tilde{X}_e, \tilde{X}_t]$) to compare methods. Each box represents one alignment method. The box indicates the lower and upper quantiles of the data, with a horizontal line at the median. The vertical line extended from the boxplot shows a 1.5 interquartile range beyond the 75th percentile or 25th percentile. The machine learning methods for alignment include linear manifold alignment (LMA) and nonlinear manifold alignment (NMA)¹⁵, manifold warping (MW)¹⁶, manifold alignment based on maximum mean discrepancy measure (MMD-MA)¹⁷, unsupervised topological alignment of single-cell multi-omics integration (UnionCom)³⁷, Single-Cell alignment using Optimal Transport (SCOT)³⁸, Manifold Aligning GAN (MAGAN)³⁹, Canonical Correlation Analysis (CCA), Reduced Rank Regression (RRR)^{5,18}, Principal Component Analysis (PCA, no alignment) and t-Distributed Stochastic Neighbor Embedding (t-SNE, no alignment)⁹.

-- Ref 1.4 Descriptions on determining hyperparameters--

Reviewer Comment	The authors may need to clearly present how the hyperparameters of the applied methods were determined given datasets.
------------------	--

Author Response	Thank the reviewer for pointing out this, which will help our manuscript clear. As requested, we elaborated on how we determined hyperparameters in the revision. We have three major hyperparameters: k is the number of nearest neighbors for building cell-cell correspondences, d is the dimension of the latent space and μ trades off the contribution between the preserving local similarity for each modality and the correspondence of the cross-modal network. First, we tried different values of k and d and found that NMA always aligns better than other methods (Fig. S13). Thus, we used $k=2$ and $d=3$ which achieve the minimum average distance among the same cells. For hyperparameter μ, we set it to be 0.5 so that the intra- and inter- modal correspondences contribute equally to the loss function because (1) we want to balance the losses from cross-modal correspondence (inter-modal) and local similarity within each modality (intra-modal), and (2) we would like to make it comparable with other methods, such as MMD-MA, UnionCom, SCOT, and MAGAN, all of which also assign equal weights to all losses from intra- and inter-modal contributions.
Excerpt From Revised Manuscript	Line 399, Page 10: “As shown on Fig. S13, we tried different values of k ($k=2, 5, 8, 10$) and d ($d=3, 5, 8, 10$) and found that as k and d grow, the distances of aligned cells did not change much and NMA always outperforms others. Thus, we used $k=2$ and $d=3$ which achieve the minimum average distance among the same cells. The parameter μ trades off the contribution between the preserving local similarity for each modality (intra-modal) and the correspondence of the cross-modal network (inter-modal). We used $\mu = 0.5$ to balance two losses. Moreover, this also makes our alignment comparable with other methods, such as MMD-MA, UnionCom, SCOT, and MAGAN, all of which also assign equal weights to all losses from intra- and inter- modal contributions.”

A

B

Figure S13 (A) Boxplots show the pairwise cell distance (Euclidean Distance) after alignment on the latent space for 3654 neuronal cells in the mouse visual cortex for different choices of nearest neighbors in manifold learning methods ranging from 2 to 10 (also see Fig 1B). The cell coordinates on the latent space are standardized per cell (i.e., each row of $\tilde{X} = [\tilde{X}_e, \tilde{X}_t]$) for comparison across methods. Each box represents one alignment method. The box indicates the lower and upper quantiles of the data, with a horizontal line at the median, the vertical line extended from the boxplot shows 1.5 interquartile range beyond the 75th percentile or 25th percentile. The machine learning methods for alignment include linear manifold alignment (LMA), nonlinear manifold alignment (NMA), manifold warping (MW), Canonical Correlation Analysis (CCA), and Principal Component Analysis (PCA, no alignment). **(B)** Boxplots show the pairwise cell distance (Euclidean Distance) after alignment on the latent space for 3654 neuronal cells in the mouse visual cortex for different choices of latent space dimensions ranging from 3 to 10.

Reviewer 2

-- Ref 2.1 Summary and strengths --

Reviewer Comment	This paper applies manifold alignment algorithms to learn a cross-modal embedding combining gene expression and electrophysiological data of single neuronal cells in the mouse brain from Brain Initiative. The dataset is obtained from a method called Patch-seq that simultaneously measures the transcriptome and electrophysiological features of a single cell. Therefore, this dataset has 1-1 correspondence between the cells. The alignment methods explored in this work include linear manifold alignment (LM), nonlinear manifold alignment (NMA), manifold warping (MW), and manifold alignment based on maximum mean discrepancy (MMD-MA). Other methods like Canonical Correlation Analysis, Reduced Rank Regression (RRR), Principal Component Analysis (PCA, without alignment), and t-Distributed Stochastic Neighbor Embedding (t-SNE, without alignment) are also used. The paper reports best alignment results for the NMA method using the euclidean distance metric between true cell pairs across gene expression and electrophysiological data modalities. NMA is also able to preserve the known cell types in the aligned space, which is measured by using silhouette scores. Finally, the paper presents results of downstream analysis of the aligned datasets to demonstrate the biological significance of performing such alignments. Overall, the paper makes a reasonable case for performing alignments across modalities in single-cell datasets. However, it requires robust evaluation metrics and additional experiments to strongly support its claim regarding the accuracy of the alignment algorithms and the usefulness of the information captured by the alignment. Please see the
------------------	---

	detailed comments below. Strengths: ++ The paper presents a useful application of the manifold alignment algorithms for analyzing the output of the Patch-seq datasets obtained from the mouse brain. Previous methods working with this dataset have focused on the cross-modal prediction task (Gala et al., 2021), while this paper presents results for cross-modal alignment of the transcriptome and electrophysiological datasets of single-cell. ++ The paper includes 8 different methods to show that overall NMA provides the best alignment performance and then proceeds to use the results from it for downstream analysis. ++ The paper uses silhouette score metric to show that the cell types are preserved in the NMA aligned space. Furthermore, one can find trajectories in the transcriptome data and by mapping the electrophysiological data modality to it, can uncover the same for it. ++ The unsupervised clustering on the aligned space reveals biological mechanisms through GO and GRN analysis
Author Response	Thank the reviewer for the great summary and acknowledging our strengths. We have addressed the weaknesses as follows.

-- Ref 2.1 Comparison with MMD-MA --

Reviewer Comment	The NMA method uses the 1-1 correspondence information when aligning the datasets and gets better performance over MMD-MA, which is an unsupervised algorithm that does not use any correspondence information for alignment. Therefore, the comparison might be a bit misleading.
Author Response	Thank the reviewer for the insightful comment. We agree that NMA uses the 1-1 cell correspondences that is a unique feature of Patch-seq which simultaneously measures gene expression and electrophysiological data of same cells. In addition to MMD-MA, we also benchmarked other recent alignment methods in the revision including supervised based MAGAN and found that NMA outperforms them in both pairwise and cell distance (Euclidean Distance) and fractions of samples closer than the true match (FOSCTTM) (Fig. 1B and Fig. S1). Also, as suggested by Ref. 3.1, we performed a semi-supervised learning test on those methods which only used 1-1 correspondence information of 50% of 3654 neuronal cells (aspiny) in

	the mouse visual cortex to infer the correspondence of other 50% cells from alignment. As shown on Fig. S11, NMA also outperforms MMD-MA and others except UnionCom. This suggests potential usefulness of NMA for aligning single-cell multi-modal data using partial correspondence information. Furthermore, given that multi-modal data may be unavailable for all cells in certain cases (e.g., morphological data is only available for a fraction of cells in Patch-seq), we included this semi-supervised analysis into the Discussion and also added a discussion to point out potential usefulness of unsupervised methods including MMD-MA, SCOT and UnionCom for single-cell multi-modal data alignment, especially when some modalities are unavailable for all cells.
Excerpt From Revised Manuscript	Line 304 Page 8: “Our nonlinear manifold alignment (NMA) uses the known cell correspondence information (1-to-1 from same cells) that is a unique feature of Patch-seq which simultaneously measures gene expression and electrophysiological data of same cells. Thus, it is expected that NMA outperforms the unsupervised alignment methods such as SCOT, MMD-MA and UnionCom. Those unsupervised methods do not need any prior knowledge on cell correspondences for alignment. Instead, they infer such correspondences in the alignment. Thus, they can be useful for aligning single-cell multi-modal data when some modalities are unavailable for all cells (e.g., morphological data is only available for a fraction of cells in Patch-seq). Also, we performed a semi-supervised learning test for evaluating the alignment performance of NMA and other methods using partial cell correspondence information. We only used 1-to-1 correspondence information of 50% of 3654 neuronal cells in the mouse visual cortex to infer the correspondence of other 50% cells from alignment. As shown on Fig. S11, NMA still outperforms others except UnionCom, suggesting potential usefulness of NMA for aligning single-cell multi-modal data using partial correspondence information.”

Figure S11 Boxplots show the pairwise cell Euclidean distance after 50% semi-supervised NMA alignment on the latent space for 3654 neuronal cells (aspiny) in the mouse visual cortex. Among the proposed methods, only UnionCom (average 0.280) outperforms NMA (average 0.587 distance)

-- Ref 2.2 Latent dimension size --

Reviewer Comment	On a similar note, the paper chose a lower dimension size of $d=3$ to align the modalities, however, one may argue that the best alignment for some of the other methods may be in a different lower-dimensional space (say $d=5$). One can use PCA to project the best alignment in $d=3$ space for visualization but restricting the alignment space to $d=3$ may not be ideal for all the methods.
Author Response	Thank the reviewer for pointing out this. We have considered the latent dimensions for achieving best alignments. We tried different values for the lower dimension size d of the latent space. As Fig. S13B shows, when d grows, the alignment distances of each method increase and NMA always outperforms others including PCA. Thus, we selected $d=3$ for achieving the minimum alignment distances.

Figure S13 (B) Boxplots show the pairwise cell distance (Euclidean Distance) after alignment on the latent space for 3654 neuronal cells in the mouse visual cortex for different choices of latent space dimensions ranging from 3 to 10. The cell coordinates on the latent space are standardized per cell (i.e., each row of $\tilde{X} = [\tilde{X}_e, \tilde{X}_t]$) for comparison across methods. Each box represents one alignment method. The box indicates the lower and upper quantiles of the data, with a horizontal line at the median, the vertical line extended from the boxplot shows 1.5 interquartile range beyond the 75th percentile or 25th percentile. The machine learning methods for alignment include linear manifold alignment (LM), nonlinear manifold alignment (NMA), manifold warping (MW), Canonical Correlation Analysis (CCA), Principal Component Analysis (PCA, no alignment), and reduced rank regression (RRR).

-- Ref 2.3 Descriptions on exploring hyperparameters --

Reviewer Comment	Furthermore, for all the methods applied in the paper, was a reasonable number of hyperparameters explored before picking the final best-performing results? If so, these details should be presented. The paper explores the k hyperparameter of the NMA algorithm. It is unclear if something similar was done for mu and d.
---

Author Response	Thank the reviewer. As requested, we elaborated on how we determined hyperparameters in the revision. We have three major hyperparameters: k is the number of nearest neighbors for building cell-cell correspondences, d is the dimension of the latent space and μ trades off the contribution between the preserving local similarity for each modality and the correspondence of the cross-modal network. First, we tried different values of k and d and found that NMA always aligns better than other methods (Fig. S13). Thus, we used $k=2$ and $d=3$ which achieve the minimum average distance among the same cells. For hyperparameter μ, we set it to be 0.5 so that the intra- and inter- modal correspondences contribute equally to the loss function because (1) we want to balance the losses from cross-modal correspondence (inter-modal) and local similarity within each modality (intra-modal), and (2) we would like to make it comparable with other methods, such as MMD-MA, UnionCom, SCOT, and MAGAN, all of which also assign equal weights to all losses from intra- and inter-modal contributions.
Excerpt From Revised Manuscript	Line 399, Page 10: “As shown on Fig. S13, we tried different values of k ($k=2, 5, 8, 10$) and d ($d=3, 5, 8, 10$) and found that as k and d grow, the distances of aligned cells did not change much and NMA always outperforms others. Thus, we used $k=2$ and $d=3$ which achieve the minimum average distance among the same cells. The parameter μ trades off the contribution between the preserving local similarity for each modality (intra-modal) and the correspondence of the cross-modal network (inter-modal). We used $\mu = 0.5$ to balance two losses. Moreover, this also makes our alignment comparable with other methods, such as MMD-MA, UnionCom, SCOT, and MAGAN, all of which also assign equal weights to all losses from intra- and inter- modal contributions.”

A**B**
Figure S13 (A) Boxplots show the pairwise cell distance (Euclidean Distance) after alignment on the latent space for 3654 neuronal cells in the mouse visual cortex for different choices of nearest neighbors in manifold learning methods ranging from 2 to 10 (also see Fig 1B). The cell coordinates on the latent space are standardized per cell (i.e., each row of $\tilde{X} = [\tilde{X}_e, \tilde{X}_t]$) for comparison across methods. Each box represents one alignment method. The box indicates the lower and upper quantiles of the data, with a horizontal line at the median, the vertical line extended from the boxplot shows 1.5 interquartile range beyond the 75th percentile or 25th percentile. The machine learning methods for alignment include linear manifold alignment (LMA), nonlinear manifold alignment (NMA), manifold warping (MW), Canonical Correlation Analysis (CCA), and Principal Component Analysis (PCA, no alignment). **(B)** Boxplots show the pairwise cell distance (Euclidean Distance) after alignment on the latent space for 3654 neuronal cells in the mouse visual cortex for different choices of latent space dimensions ranging from 3 to 10.

-- Ref 2.4 MAGAN --

Reviewer Comment	The paper does not mention MAGAN (Amodio et al, 2018) and UnionCom (Cao et al., 2020) which are also manifold alignment methods, with the former using a deep learning architecture and the latter a variation of Generalized Unsupervised Manifold Alignment (Cui et al. 2014)
Author Response	Thank the reviewer for suggesting MAGAN. As suggested, we have applied several additional manifold learning methods UnionCom, SCOT and MAGAN to the mouse visual cortex dataset. We found that NMA still outperforms those methods in both pairwise and cell distance (Euclidean Distance) and fractions of samples closer than the true match (FOSCTTM) (suggested by Ref 2.6) (Fig. 1B and Fig. S1). We included this comparative analysis into the revision and cited MAGAN as well.

Figure 1B (top) and **Figure S1** (bottom) show the pairwise cell distance (Euclidean Distance) and fractions of samples closer than the true match (FOSCTTM), respectively, after alignment on the latent space for 3654 neuronal cells (aspiny) in the mouse visual cortex. The cell coordinates on the latent space are standardized per cell (i.e., each row of $\tilde{X} = [\tilde{X}_e, \tilde{X}_t]$) to compare methods. Each box represents one alignment method. The box indicates the lower and upper quartiles of the data, with a horizontal line at the median. The vertical line extended from the boxplot shows a 1.5 interquartile range beyond the 75th percentile or 25th percentile. The machine learning methods for alignment include linear manifold alignment (LMA) and nonlinear manifold alignment (NMA)¹⁵, manifold warping (MW)¹⁶, manifold alignment based on maximum mean discrepancy measure (MMD-MA)¹⁷, unsupervised topological alignment of single-cell multi-omics integration (UnionCom)³⁷,

Single-Cell alignment using Optimal Transport (SCOT)³⁸, Manifold Aligning GAN (MAGAN)³⁹, Canonical Correlation Analysis (CCA), Reduced Rank Regression (RRR)^{5,18}, Principal Component Analysis (PCA, no alignment) and t-Distributed Stochastic Neighbor Embedding (t-SNE, no alignment)⁹.

-- Ref 2.5 Single-cell multi-modal data --

Reviewer Comment	Is there a specific reason why the focus is on manifold alignment methods since multiple methods have been proposed for alignment of data modalities with 1-1 supervision like Seurat V4 (Hao et al.), LIGER (Welch et al. 2019), etc.? If so, the paper must elaborate on that.
Author Response	This is a great point. Many existing alignment methods have been applied to align single-cell multi-omics data such as scRNA-seq and scATAC-seq, but not to recent Patch-seq single-cell multi-modal data including non-omic modalities such as electrophysiology. To this end, this paper aims to fill this gap and benchmarks those methods for aligning Patch-seq transcriptomic and electrophysiological data of single cells in mouse brains. We compared both manifold alignment methods and non-manifold based methods such as CCA by Seurat, but found that nonlinear manifold alignment (NMA) outperforms others. This suggests potential high nonlinear relationships among gene expression and electrophysiology, likely found by manifolds. So, we selected NMA for downstream analyses to further identify cross-modal cell clusters and predict genes and regulatory networks for electrophysiological features in the clusters. We also agree that NMA takes advantage of the 1-1 cell correspondences as a priori that is a unique feature of Patch-seq which simultaneously measures gene expression and electrophysiological data of same cells. We elaborated on manifold alignment in the revised manuscript as below.
Excerpt From Revised Manuscript	Line 89, Page 3: “The alignment methods have been previously used to align single-cell multi-omics data, e.g., scRNA-seq and scATAC-seq. Mathematically, these methods align multi-omics data of single cells and project the cells from different omics onto a latent space (e.g., co-embedding). The cells aligned on the latent space likely form certain cell clusters and share biological mechanisms, e.g., gene regulation from aligning scRNA-seq and scATAC-seq. For instance, the linear alignment methods such as canonical correlation analysis (CCA) (e.g., Seurat 22) and RRR decompose single-cell data matrices of different omics (e.g., genes and regulatory elements across cells) to find lower-dimensional representative factors across omics. Those factors can be used to cluster cells and find the clusters’ omics activities. As nonlinear alignment methods, MAGAN applies manifold alignment to match cells from single-cell multi-omics datasets using generative adversarial networks. It empirically requires biological manifolds (e.g., known cell types) to build the cell correspondences across omics for better alignment. Recently, UnionCom

extends the generalized unsupervised manifold alignment (GUMA) to embed cells from each omics onto a lower-dimensional latent space (via kNN) and then match cross-omics spaces to align cells. Besides, Maximum Mean Discrepancy-Manifold Alignment (MMD-MA) embeds the latent spaces onto a common reproducing kernel Hilbert space by minimizing the MMD across omics. Also, SCOT uses the optimal transport technique to project one modality onto the space of another while preserving the local neighborhood of geometry from the modality. Although those methods have been shown that aligned cells have somehow specific omics activities, they have not been widely applied and tested to align additional modalities such as gene expression vs. electrophysiology, which typically have complex and likely nonlinear cross-modal relationships (more nonlinear than cross-omics).”

-- Ref 2.6 FOSCTTM to evaluate alignment performance --

Reviewer Comment	The evaluation metric for evaluating the alignment performance is euclidean distance. Technically, the true pair might be distant from each other in the aligned space, but the alignment still works if a sample point is closer to its true match compared to other sample points. Therefore, a metric like fractions of samples closer than the true match (FOSCTTM), used for performance evaluation of MMD-MA and UnionCom might be better to evaluate the performance. It might also be worth reporting the Label Transfer Accuracy metric (used in UnionCom paper).
Author Response	Thank the reviewer. As suggested, we used FOSCTTM to evaluate the alignment performance and found that NMA still outperforms other methods. In particular, we calculated the FOSCTTM as follows. For each cell in the electrophysiological data, we first find its true match in the gene expression data, and then rank all other cells on the aligned latent space based on their distances from x , and then compute the fraction of cells that are closer than the true match. As shown on a new Fig. S1, NMA has significantly smaller FOSCTTM distances than others. We added this analysis into the revision.
Excerpt From Revised Manuscript	Figure S1 Boxplots show the pairwise cell Mean FOSCTTM Score after alignment on the latent space for 3654 neuronal cells (aspiny) in the mouse visual cortex (Methods). FOSCTTM: fractions of samples closer than the true match. The cell coordinates on the latent space are standardized per cell (i.e., each row of) to compare methods. Each box represents one alignment method. The box indicates the lower and upper quantiles of the data, with a horizontal line at the median. The vertical line extended from the boxplot shows a 1.5 interquartile range beyond the 75th percentile or 25th percentile. The machine learning methods for alignment include linear manifold alignment (LMA), nonlinear manifold alignment (NMA), manifold warping (MW), Canonical Correlation Analysis (CCA), Reduced Rank Regression (RRR), Principal Component Analysis (PCA, no alignment), t-SNE (Stochastic

Neighbor Embedding, no alignment), MMD-MA (Manifold Alignment with maximum mean discrepancy measurement), unsupervised topological alignment of single-cell multi-omics integration (UnionCom), Single-Cell alignment using Optimal Transport (SCOT), and Manifold Aligning GAN (MAGAN).

Line 421, Page 11: “In addition to the pairwise distances of cells on the common latent space, we also used the metric, fractions of samples closer than the true match (FOSCTTM) 18 for evaluation. In particular, we calculated the FOSCTTM score of aligned cells as follows. For each cell in the electrophysiological data, we first find its true match in the gene expression data, and then rank all other cells on the aligned latent space based on their distances from x, and then compute the fraction of cells that are closer than the true match.”

-- Ref 2.7 Downstream Analysis for PCA --

Reviewer Comment	The significance of the downstream analysis of NMA space is hard to gauge. It would help to show that if one was to perform a similar analysis on the individual PCA of the two modalities (without alignment), they will not be as informative. Or one modality may turn out to be informative (like transcriptome) but the other will require alignment to be as informative. It is most likely the case but there are no experiments to highlight this. One could also do this to compare other alignment methods.
Author Response	Thank the reviewer for this insightful point. As suggested, we used the same clustering method to cluster the cells using single modality (gene expression or electrophysiology) on the PCA space without alignment. We found that those single-modal cell clusters are not so consistent with t-types as cross-modal clusters after alignment. For instance, no single-modal clusters have over 70% of Vip-type, Lamp5-type and Sst-type cells, whereas cross-modal Cluster 4 has ~83.3% Lamp5-type cells, Cluster 3 has ~86.6% Sst-type cells and Cluster 1 has ~79.1% Vip cells. Thus, this suggests that multi-modal alignment helps clustering together the cells from the same types, compared to using single modality only such as electrophysiology. We added this analysis into the revision.
Excerpt From Revised Manuscript	Line 198, Page 5: “Those cell clusters are cross-modal clusters since they are formed after aligning their gene expression and electrophysiological data. As expected, they are highly in accordance with t-types (Fig. S7). For example, Cluster 4 has ~83.3% Lamp5-type cells (373/448 cells), Cluster 2 has ~77.6% Pvalb-type cells (558/719 cells), Cluster 3 has ~86.6% Sst-type cells (1339/1546 cells) and Cluster 1 has ~79.1% Vip cells (541/684 cells). Besides, Clusters 1 and 5 include ~55.8% Serpinfl cells (24/43) and ~60.7% Sncg cells

(84/214), respectively. Moreover, we used the same clustering method to cluster the cells using single modality (gene expression or electrophysiology) on the PCA space without alignment. We found that those single-modal cell clusters are not so consistent with t-types as cross-modal clusters after alignment. For instance, by using electrophysiological data only, we found that the cell clusters include 57.8% Lamp5-type cells, 85.1% Pvalb-type cells, 65.1% Serpinf1-type cells, 63.1% Sncg-type cells, 49.5% Sst-type cells, and 60.8% Vip-type cells. Using gene expression data only, the cell clusters have 68.9% Lamp5-type cells, 54.4% Pvalb-type cells, 55.8% Serpinf1-type cells, 67.3% Sncg-type cells, 45.2% Sst-type cells, and 65.2% Vip-type cells. Thus, no single-modal clusters have over 70% of Vip-type, Lamp5-type and Sst-type cells. This suggests that multi-modal alignment is not driven by single modalities and also helps clustering together the cells from the same types.”

-- Ref 2.8 Compare to Gala et al. --

Reviewer Comment	It would also be useful to discuss how do the cross-modality prediction results compare to the one reported in Gala et al. (2021)
Author Response	Thank the reviewer for suggesting the recent paper (Gala et al., Nature Computational Science 2021). This paper did not use the alignment method to align multi-modal data onto the same latent space. Instead, they used coupled autoencoders to project gene expression and electrophysiological features onto two separate latent spaces. Also, since the autoencoder model is based on deep neural networks, their method is computationally intensive such as involving tuning many hyperparameters. Moreover, they used known genes in cell type specific paracrine signaling pathways for predicting electrophysiological features, whereas our prediction is more general without needing any prior biological knowledge on genes (i.e., using differentially expressed genes of cross-modal cell clusters for prediction). However, we admit that the coupled autoencoder modeling in Gala et al. 2021 is innovative and such deep-learning based models might be able to help improve multi-modal data alignment in future. Thus, we included a brief discussion in the Discussion of the revised manuscript.
Excerpt From Revised Manuscript	Line 317, Page 8: “Furthermore, deep learning models have been proposed for cross-modal prediction. For example, a coupled autoencoder model ¹² was proposed to align Patch-seq data to project gene expression and electrophysiological features onto two separate latent spaces. Although computationally intensive such as involving tuning many hyperparameters, given relatively large sample sizes from single cell data, such deep-learning based models might be able to help improve multi-modal data alignment in future.”

-- Ref 2.9 literature review on other methods --

Reviewer Comment	Brief descriptions of some of the other manifold alignment methods applied would be very useful.
Author Response	Thank the reviewer. We added a literature review on the alignment methods as follows.
Excerpt From Revised Manuscript	Line 76, Page 2: “We have applied and benchmarked multiple existing machine learning methods to align the single cells in the mouse brain using their gene expression and electrophysiological data (Methods, Fig. 1A). In particular, we focused on two major brain regions, mouse visual cortex and motor cortex, and used the latest Patch-seq data from Allen Brain Atlas in the BRAIN Initiative ^{5,13,14} (Methods). The machine learning methods for alignment include linear manifold alignment (LMA) and nonlinear manifold alignment (NMA) ¹⁵, manifold warping (MW) ¹⁶, manifold alignment based on maximum mean discrepancy measure (MMD-MA) ¹⁷, unsupervised topological alignment of single-cell multi-omics integration (UnionCom) ¹⁸, Single-Cell alignment using Optimal Transport (SCOT) ¹⁹, Manifold Aligning GAN (MAGAN) ²⁰, Canonical Correlation Analysis (CCA), Reduced Rank Regression (RRR) ^{5,21}, Principal Component Analysis (PCA, no alignment) and t-Distributed Stochastic Neighbor Embedding (t-SNE, no alignment) ⁹. The alignment methods have been previously used to align single-cell multi-omics data, e.g., scRNA-seq and scATAC-seq. Mathematically, these methods align multi-omics data of single cells and project the cells from different omics onto a latent space (e.g., co-embedding). The cells aligned on the latent space likely form certain cell clusters and share biological mechanisms, e.g., gene regulation from aligning scRNA-seq and scATAC-seq. For instance, the linear alignment methods such as canonical correlation analysis (CCA) (e.g., Seurat ²²) and RRR decompose single-cell data matrices of different omics (e.g., genes and regulatory elements across cells) to find lower-dimensional representative factors across omics. Those factors can be used to cluster cells and find the clusters’ omics activities. As nonlinear alignment methods, MAGAN applies manifold alignment to match cells from single-cell multi-omics datasets using generative adversarial networks. It empirically requires biological manifolds (e.g., known cell types) to build the cell correspondences across omics for better alignment. Recently, UnionCom extends the generalized unsupervised manifold alignment (GUMA) to embed cells from each omics onto a lower-dimensional latent space (via kNN) and then match cross-omics spaces to align cells. Besides, Maximum Mean Discrepancy-Manifold Alignment (MMD-MA) embeds the latent spaces onto a common reproducing kernel Hilbert space by minimizing the MMD across omics. Also, SCOT uses the optimal transport technique to project one modality onto the space of another while preserving the local neighborhood of geometry from the modality. Although those methods have been shown that aligned cells have somehow specific omics activities, they have not been widely applied and tested to align additional modalities such as gene expression vs.

electrophysiology, which typically have complex and likely nonlinear cross-modal relationships (more nonlinear than cross-omics).”

-- Ref 2.10 Reporting high correlations in prediction --

Reviewer Comment	It is likely that I missed something in the description and would like to clarify my confusion regarding the cross-modal prediction result. Given that the NMA method explicitly matches the features across the two datasets, it is unclear what is the significance of reporting high correlation when regression is performed using the components of the NMA space? Is it not to be expected?
Author Response	We agree with the reviewer that reporting the correlations of NMA components from the biplot analysis is unnecessary since the high correlations between aligned components are expected. Thus, we removed those R^2 values in the revision. However, we still keep reporting the R^2 values from predicting electrophysiological features from differentially expressed genes of cross-modal clusters, revealing the predictability of gene expression for electrophysiology.

Reviewer 3

-- Ref 3.1 Semi-supervised method --

Reviewer Comment	The biggest concern is with the comparison measures. The NMA method is purely supervised in the sense the cross-correspondence matrix is given a priori. Thus it is expected for the same cell distances in the final embedding to be small. Thus, the boxplot fig 1B is not relevant. You can achieve a perfect score (almost 0 distance between the corresponding cells, by just setting the u parameter close to 0). For instance, MMD-MA is unsupervised, therefore it should work worse for this kind of comparison against a supervised method. Thus the comparisons between the supervised and unsupervised methods are unfair. An alternative approach would be to perform the algorithm using just a small set of cross-correspondence relationships, in a more realistic semi-supervised fashion and compare it to (Cao, K., 2020, reference below). You could do this by measuring the Euclidean distances of the same cells on the latent space, but for the observations not considered in the construction of the cross-correspondence matrix.
------------------	---

Author Response	Thank the reviewer for this great suggestion. We agree that NMA takes advantage of the 1-1 cell correspondences as a priori that is a unique feature of Patch-seq which simultaneously measures gene expression and electrophysiological data of same cells. However, as suggested by the reviewer, we performed a semi-supervised learning test on NMA only using 1-1 correspondence information of 50% of 3654 neuronal cells (aspiny) in the mouse visual cortex to infer the correspondence of other 50% cells from alignment. As shown on Fig. S11, NMA also outperforms MMD-MA and others except UnionCom. This suggests potential usefulness of NMA for aligning single-cell multi-modal data using partial correspondence information. Furthermore, given that multi-modal data may be unavailable for all cells in certain cases (e.g., morphological data is only available for a fraction of cells in Patch-seq), we included this semi-supervised analysis into the Discussion and also added a discussion to point out potential usefulness of unsupervised methods including MMD-MA, SCOT and UnionCom for single-cell multi-modal data alignment, especially when some modalities are unavailable for all cells. Finally, we set μ to be 0.5 in NMA so that the intra- and inter- modal correspondences contribute equally to the loss function because (1) we want to balance the losses from cross-modal correspondence (inter-modal) and local similarity within each modality (intra-modal), and (2) we would like to make it comparable with other methods, such as MMD-MA, UnionCom, SCOT, and MAGAN, all of which also assign equal weights to all losses from intra- and inter- modal contributions.
Excerpt From Revised Manuscript	Line 304, Page 8: “Our nonlinear manifold alignment (NMA) uses the known cell correspondence information (1-to-1 from same cells) that is a unique feature of Patch-seq which simultaneously measures gene expression and electrophysiological data of same cells. Thus, it is expected that NMA outperforms the unsupervised alignment methods such as SCOT, MMD-MA and UnionCom. Those unsupervised methods do not need any prior knowledge on cell correspondences for alignment. Instead, they infer such correspondences in the alignment. Thus, they can be useful for aligning single-cell multi-modal data when some modalities are unavailable for all cells (e.g., morphological data is only available for a fraction of cells in Patch-seq). Also, we performed a semi-supervised learning test for evaluating the alignment performance of NMA and other methods using partial cell correspondence information. We only used 1-to-1 correspondence information of 50% of 3654 neuronal cells in the mouse visual cortex to infer the correspondence of other 50% cells from alignment. As shown on Fig. S11, NMA still outperforms others except UnionCom, suggesting potential usefulness of NMA for aligning single-cell multi-modal data using partial correspondence information.”

Figure S11 Boxplots show the pairwise cell Euclidean distance after 50% semi-supervised NMA alignment on the latent space for 3654 neuronal cells (aspiny) in the mouse visual cortex. Among the proposed methods, only UnionCom (average 0.280) outperforms NMA (average 0.587 distance)

Line 404, Page 11: “The parameter μ trades off the contribution between the preserving local similarity for each modality (intra-modal) and the correspondence of the cross-modal network (inter-modal). We used $\mu=0.5$ to balance two losses. Moreover, this also makes our alignment comparable with other methods, such as MMD-MA, UnionCom, SCOT, and MAGAN, all of which also assign equal weights to all losses from intra- and inter- modal contributions.”

-- Ref 3.2 Alignment not driven by single modalities --

Reviewer
Comment

There are also some remaining questions about the approach. Does the construction of the similarity matrices for each modality use a kernel? Or is it just a 1-0 matrix indicating the nearest neighbors?. More explanation about this is required, especially since what might happen in the objective function is that the similarities of some modality might be much stronger than in the other. And in such a scenario, the objective function might just prefer to take one of the modalities into account, and in junction with the cross-correspondence similarities (which might be the strongest part of the objective function), the method would just create an embedding of the transcriptomics (or the electrophysiology) and then “match” the cells, not taking into account the electrophysiology features (or the transcriptomics). In this sense, the separate embeddings of each domain before alignment are needed to assess the impact of the alignment (see some examples of this in

	the suggested references below). Liu, J., Huang, Y., Singh, R., Vert, J. P., & Noble, W. S. (2019). Jointly embedding multiple single-cell omics measurements. BioRxiv, 644310. Cao, K., Bai, X., Hong, Y., & Wan, L. (2020). Unsupervised topological alignment for single-cell multi-omics integration. Bioinformatics Amodio, M., & Krishnaswamy, S. (2018, July). MAGAN: Aligning biological manifolds. In International Conference on Machine Learning Cui, Z., Chang, H., Shan, S., & Chen, X. (2014). Generalized unsupervised manifold alignment. Advances in Neural Information Processing Systems, 27, 2429-2437.
Author Response	Thank the reviewer for the suggestion. We used the 1-0 matrix as the similarity matrix to indicate the k-nearest neighbors, in which the ith row and jth column is value 1 if cell j is a k-nearest neighbor of cell i. We clarified this in the revision. As suggested, we used the same clustering method to cluster the cells using single modality (gene expression or electrophysiology) on the PCA space without alignment. We found that those single-modal cell clusters are not so consistent with t-types as cross-modal clusters after alignment. For instance, no single-modal clusters have over 70% of Vip-type, Lamp5-type and Sst-type cells, whereas cross-modal Cluster 4 has ~83.3% Lamp5-type cells, Cluster 3 has ~86.6% Sst-type cells and Cluster 1 has ~79.1% Vip cells. Thus, this suggests that multi-modal alignment is not driven by single modalities and also helps clustering together the cells from the same types. We added this analysis in the revision. Also, we cited those papers in the revised manuscript. Thanks again.
Excerpt From Revised Manuscript	Line 192, Page 5: “Those cell clusters are cross-modal clusters since they are formed after aligning their gene expression and electrophysiological data. As expected, they are highly in accordance with t-types (Fig. S7). For example, Cluster 4 has ~83.3% Lamp5-type cells (373/448 cells), Cluster 2 has ~77.6% Pvalb-type cells (558/719 cells), Cluster 3 has ~86.6% Sst-type cells (1339/1546 cells) and Cluster 1 has ~79.1% Vip cells (541/684 cells). Besides, Clusters 1 and 5 include ~55.8% Serpinf1 cells (24/43) and ~60.7% Sncg cells (84/214), respectively. Moreover, we used the same clustering method to cluster the cells using single modality (gene expression or electrophysiology) on the PCA space without alignment. We found that those single-modal cell clusters are not so consistent with t-types as cross-modal clusters after alignment. For instance, by using electrophysiological data only, we found that the cell clusters include 57.8% Lamp5-type cells, 85.1% Pvalb-type cells, 65.1% Serpinf1-type cells, 63.1% Sncg-type cells, 49.5% Sst-type cells, and 60.8% Vip-type cells. Using gene expression data only, the cell clusters have

68.9% Lamp5-type cells, 54.4% Pvalb-type cells, 55.8% Serpinf1-type cells, 67.3% Sncg-type cells, 45.2% Sst-type cells, and 65.2% Vip-type cells. Thus, no single-modal clusters have over 70% of Vip-type, Lamp5-type and Sst-type cells. This suggests that multi-modal alignment is not driven by single modalities and also helps clustering together the cells from the same types.”

-- Ref 3.3 Additional clustering approaches --

Reviewer Comment	Regarding this previous comment, the clustering comparison might be biased. The final embedding might prefer the transcriptomics domain keeping a good separation between t-types. Given that the paper aims to do benchmarking, the comparison should be more extensive. By using different clustering approaches and performance measures, this would give a more general idea of the robustness of different methods.																																																																																							
Author Response	As suggested, we also applied K-medoid and hierarchical clustering methods, two other popular clustering methods to cluster aligned cells into cross-modal cell clusters. We also found that those clusters highly overlap and also are consistent with t-types. This suggests a robustness of clustering cross-modal aligned cells. We also included this analysis in the revision.																																																																																							
Excerpt From Revised Manuscript	Line 208, Page 6: “Furthermore, in addition to GMM, we also used K-medoid and Hierarchical clustering methods to cluster aligned cells and cross-modal cell clusters. Those cross-modal clusters highly overlap with t-types as well (Fig. S8), suggesting the robustness of clustering cross-modal aligned cells. K-medoid clusters together 90.2% Lamp5-type cells, 96.6% Pvalb-type cells, 55.8% Serpinf1-type cells, 61.7% Sncg-type cells, 96.5% Sst-type cells, and 75.7% Vip-type cells. Hierarchical clustering clusters together 79.9% Lamp5-type cells, 98.6% Pvalb-type cells, 83.7% Serpinf1-type cells, 57.4% Sncg-type cells, 94.6% Sst-type cells, and 94.9% Vip-type cells.”   <caption>Data for Figure S8: Numbers of overlapped cells</caption>   t-type K-medoid Hierarchical Clustering   Cluster1 Cluster2 Cluster3 Cluster4 Cluster5 Cluster1 Cluster2 Cluster3 Cluster4 Cluster5     Lamp5 ~100 ~100 ~100 ~100 ~100 ~100 ~100 ~100 ~100 ~100   Pvalb ~100 ~1000 ~100 ~100 ~100 ~100 ~1000 ~100 ~100 ~100   Serpinf1 ~100 ~100 ~100 ~100 ~100 ~100 ~100 ~100 ~100 ~100   Sncg ~100 ~100 ~100 ~100 ~100 ~100 ~100 ~100 ~100 ~100   Sst ~100 ~100 ~100 ~100 ~1400 ~100 ~100 ~100 ~100 ~1400   Vip ~100 ~100 ~100 ~100 ~100 ~100 ~100 ~100 ~100 ~100     Figure S8 Numbers of overlapped cells between t-types and cross-modal cell clusters by Hierarchical Clustering (Right) and K-medoid (Left) from the latent	t-type	K-medoid					Hierarchical Clustering					Cluster1	Cluster2	Cluster3	Cluster4	Cluster5	Cluster1	Cluster2	Cluster3	Cluster4	Cluster5	Lamp5	~100	~100	~100	~100	~100	~100	~100	~100	~100	~100	Pvalb	~100	~1000	~100	~100	~100	~100	~1000	~100	~100	~100	Serpinf1	~100	~100	~100	~100	~100	~100	~100	~100	~100	~100	Sncg	~100	~100	~100	~100	~100	~100	~100	~100	~100	~100	Sst	~100	~100	~100	~100	~1400	~100	~100	~100	~100	~1400	Vip	~100	~100	~100	~100	~100	~100	~100	~100	~100	~100
t-type	K-medoid					Hierarchical Clustering																																																																																		
	Cluster1	Cluster2	Cluster3	Cluster4	Cluster5	Cluster1	Cluster2	Cluster3	Cluster4	Cluster5																																																																														
Lamp5	~100	~100	~100	~100	~100	~100	~100	~100	~100	~100																																																																														
Pvalb	~100	~1000	~100	~100	~100	~100	~1000	~100	~100	~100																																																																														
Serpinf1	~100	~100	~100	~100	~100	~100	~100	~100	~100	~100																																																																														
Sncg	~100	~100	~100	~100	~100	~100	~100	~100	~100	~100																																																																														
Sst	~100	~100	~100	~100	~1400	~100	~100	~100	~100	~1400																																																																														
Vip	~100	~100	~100	~100	~100	~100	~100	~100	~100	~100																																																																														

space after nonlinear manifold alignment of single cells in the mouse visual cortex. The dot size and color correspond to the number of shared cells.

-- Ref 3.4 PHATE to visualize transcriptomic trajectory --

Reviewer Comment	When the authors claimed t-types trajectories are not recovered by other methods, did you try PHATE in the transcriptomic domain? PHATE often does well at visualizing trajectory data. Is there a particular reason to use the Laplacian eigenmaps objective function for each modality? It could be interesting to use PHATE or tsne/umap, since those often work better.
Author Response	We appreciate this suggestion. We applied PHATE, tsne and umap along with PCA to the transcriptomic data of 3,654 cells in the mouse visual cortex. As shown on Fig. S4, we found that they do not show any single trajectory that transitions multiple t-types as NMA (Fig. 2A). However, we included this analysis in the main text along with a new supplemental figure (Fig. S4).
Excerpt From Revised Manuscript	Line 151, Page 4: “Also, since the transcriptomic types are defined by transcriptomic data, we applied PCA,t-SNE, Umap, and PHATE³⁹ to the transcriptomic data of those cells and found that those methods do not show any single trajectory transitioning t-types (Fig. S4), unlike NMA (Fig. 2A). This suggests that NMA not only recovered t-types but also found a cross-t-type trajectory visualizing transitions across t-types...”   Figure S4 PCA, PHATE, t-SNE and U-map for visualizing 3,654 cells in the mouse visual cortex using their transcriptomic data. Points: cells. Colors: transcriptomic types (t-types).

-- Ref 3.5 Literature Review for Proposed Methods --

Reviewer Comment	If the goal is to benchmark the methods more discussion of them is required in similar detail as the authors did for NMA.
Author Response	Thank the reviewer for pointing this out. We added a literature review on all methods as follows.
Excerpt From Revised Manuscript	Line 76, Page 2: “We have applied and benchmarked multiple existing machine learning methods to align the single cells in the mouse brain using their gene expression and electrophysiological data (Methods, Fig. 1A). In particular, we focused on two major brain regions, mouse visual cortex and motor cortex, and used the latest Patch-seq data from Allen Brain Atlas in the BRAIN Initiative ^{5,13,14} (Methods). The machine learning methods for alignment include linear manifold alignment (LMA) and nonlinear manifold alignment (NMA) ¹⁵, manifold warping (MW) ¹⁶, manifold alignment based on maximum mean discrepancy measure (MMD-MA) ¹⁷, unsupervised topological alignment of single-cell multi-omics integration (UnionCom) ¹⁸, Single-Cell alignment using Optimal Transport (SCOT) ¹⁹, Manifold Aligning GAN (MAGAN) ²⁰, Canonical Correlation Analysis (CCA), Reduced Rank Regression (RRR) ^{5,21}, Principal Component Analysis (PCA, no alignment) and t-Distributed Stochastic Neighbor Embedding (t-SNE, no alignment) ⁹. The alignment methods have been previously used to align single-cell multi-omics data, e.g., scRNA-seq and scATAC-seq. Mathematically, these methods align multi-omics data of single cells and project the cells from different omics onto a latent space (e.g., co-embedding). The cells aligned on the latent space likely form certain cell clusters and share biological mechanisms, e.g., gene regulation from aligning scRNA-seq and scATAC-seq. For instance, the linear alignment methods such as canonical correlation analysis (CCA) (e.g., Seurat ²²) and RRR decompose single-cell data matrices of different omics (e.g., genes and regulatory elements across cells) to find lower-dimensional representative factors across omics. Those factors can be used to cluster cells and find the clusters’ omics activities. As nonlinear alignment methods, MAGAN applies manifold alignment to match cells from single-cell multi-omics datasets using generative adversarial networks. It empirically requires biological manifolds (e.g., known cell types) to build the cell correspondences across omics for better alignment. Recently, UnionCom extends the generalized unsupervised manifold alignment (GUMA) to embed cells from each omics onto a lower-dimensional latent space (via kNN) and then match cross-omics spaces to align cells. Besides, Maximum Mean Discrepancy-Manifold Alignment (MMD-MA) embeds the latent spaces onto a common reproducing kernel Hilbert space by minimizing the MMD across omics. Also, SCOT uses the optimal transport technique to project one modality onto the space of another while preserving the local neighborhood of geometry from the modality. Although those methods have been shown that aligned cells have somehow specific omics activities, they have not been widely applied and tested to align additional modalities such as gene expression vs.

electrophysiology, which typically have complex and likely nonlinear cross-modal relationships (more nonlinear than cross-omics).”

Reviewers' comments:

Reviewer #1 (Remarks to the Author):

All my questions have been addressed.

Reviewer #3 (Remarks to the Author):

The authors have addressed many of my comments but there are still some major concerns that remain. In particular, the authors compared their method with UnionCom. Their results seem to indicate that UnionCom performs better than their proposed method on the experiments. So these are my remaining questions:

- 1) If UnionCom outperforms the proposed method, when would it make sense to use the authors' method?
- 2) Are the comparisons with UnionCom run for the purely unsupervised UnionCom or the unsupervised version using the same cross-correspondence as the authors' approach? If the cross-correspondence was not used, I imagine that UnionCom would do even better with it.
- 3) Does UnionCom or MAGAN recover the single trajectory for t-types, and the clusters? Or does only the proposed method recover such structures?
- 4) The comparison with UnionCom with just the boxplots of euclidean distances are insufficient, further experiments as in the original UnionCom paper would be better and more enlightening.

We have addressed the reviewer comments and provided the point-by-point responses as follows. We really appreciate the work of reviewers to help improve our manuscript.

Reviewer 3

-- Ref 3.1 Authors' method --

Reviewer Comment	1) If UnionCom outperforms the proposed method, when would it make sense to use the authors' method?
Author Response	Thank the reviewer. UnionCom outperforms other methods in the semi-supervised learning, as shown in our response to previous Ref 3.1. However, nonlinear manifold alignment (NMA) is still better than UnionCom in the supervised learning setting, i.e., known cell correspondence information which is a unique feature of BRAIN Initiative data that measures multi-modal data of same cells. Also, as responded to two other reviewers (previous Ref. 1.3 and Ref. 2.6), we evaluated those methods using UnionCom's metric, "fractions of samples closer than the true match (FOSCTTM)" (Fig. S1) and found that NMA still outperforms others. Moreover, UnionCom does not show any single trajectory over aligned cells (Fig. S4) as shown below. Thus, since this paper focuses on aligning BRAIN Initiative data, NMA is still a valuable approach for aligning single-cell multi-modal data and visualizing single cell trajectory from alignment.

-- Ref 3.2 UnionCom with cross-correspondence --

Reviewer Comment	2) Are the comparisons with UnionCom run for the purely unsupervised UnionCom or the unsupervised version using the same cross-correspondence as the authors' approach? If the cross-correspondence was not used, I imagine that UnionCom would do even better with it.
Author Response	Yes, we agree that UnionCom did a great job without using the same cross-correspondence information. It estimates the optimal matching matrix (cross-correspondence), allowing soft matching (i.e., weighting cell-cell correspondence) and then uses it to co-embed two modalities on the common latent space by KL-divergence. So, it does not provide any parameter to hard-code cross-correspondence. Thus, when we compared it with other methods, we respected this unique feature of UnionCom without providing any same cross-correspondence information. However, as we responded in last round, we have added text in Discussion acknowledging great potential of unsupervised alignment methods such as UnionCom for single cell multi-modal alignment, especially when some modalities are unavailable for all cells (e.g., morphological data is only available for a fraction of cells in Patch-seq).

Excerpt From Revised Manuscript	Line 304, Page 8: “Our nonlinear manifold alignment (NMA) uses the known cell correspondence information (1-to-1 from same cells) that is a unique feature of Patch-seq which simultaneously measures gene expression and electrophysiological data of same cells. Thus, it is expected that NMA outperforms the unsupervised alignment methods such as SCOT, MMD-MA and UnionCom. Those unsupervised methods do not need any prior knowledge on cell correspondences for alignment. Instead, they infer such correspondences in the alignment. Thus, they can be useful for aligning single-cell multi-modal data when some modalities are unavailable for all cells (e.g., morphological data is only available for a fraction of cells in Patch-seq)...”
---

-- Ref 3.3 Single trajectory by UnionCom/MAGAN --

Reviewer Comment	3) Does UnionCom or MAGAN recover the single trajectory for t-types, and the clusters? Or does only the proposed method recover such structures?
Author Response	Thank the reviewer for the great suggestion. We checked the aligned cells by UnionCom and MAGAN and did not find any single trajectory for t-types. Only NMA shows such a trajectory, implying highly nonlinear manifold relationships between gene expression and electrophysiology. We have added UnionCom and MAGAN to Fig. S4.
Excerpt From Revised Manuscript	Line 151, Page 4: “Also, since the transcriptomic types are defined by transcriptomic data, we applied PCA, t-SNE, Umap, and PHATE ³⁹ to the transcriptomic data and UnionCom, MAGAN to both modalities of those cells and found that those methods do not show any single trajectory transitioning t-types (Fig. S4), unlike NMA (Fig. 2A)...”  Figure S4 UnionCom, MAGAN, PCA, PHATE, t-SNE and U-map for visualizing 3,654 cells in the mouse visual cortex using their transcriptomic data. Points: cells. Colors: transcriptomic types (t-types).

-- Ref 3.4 Experiments in UnionCom --

Reviewer Comment	4) The comparison with UnionCom with just the boxplots of euclidean distances are insufficient, further experiments as in the original UnionCom paper would be better and more enlightening.
Author Response	We appreciate this suggestion. As also recommended by Ref 2.4, in addition to Euclidean distances, we indeed used “fractions of samples closer than the true match (FOSCTTM)” from UnionCom to evaluate the alignment performance. NMA still outperforms other methods in FOSCTTM. We calculated the FOSCTTM as follows. For each cell in the electrophysiological data, we first find its true match in the gene expression data, and then rank all other cells on the aligned latent space based on their distances from x, and then compute the fraction of cells that are closer than the true match. As shown on new Fig. S1, NMA has significantly smaller FOSCTTM distances than others. We have also added this analysis into the revision.
Excerpt From Revised Manuscript	   Figure S1 Boxplots show the pairwise cell Mean FOSCTTM Score after alignment on the latent space for 3654 neuronal cells (aspiny) in the mouse visual cortex (Methods). FOSCTTM: fractions of samples closer than the true match. The cell coordinates on the latent space are standardized per cell (i.e., each row of) to compare methods. Each box represents one alignment method. The box indicates the lower and upper quantiles of the data, with a horizontal line at the median. The vertical line extended from the boxplot shows a 1.5 interquartile range beyond the 75th percentile or 25th percentile. The machine learning methods for alignment include linear manifold

alignment (LMA), nonlinear manifold alignment (NMA), manifold warping (MW), Canonical Correlation Analysis (CCA), Reduced Rank Regression (RRR), Principal Component Analysis (PCA, no alignment), t-SNE (Stochastic Neighbor Embedding, no alignment), MMD-MA (Manifold Alignment with maximum mean discrepancy measurement), unsupervised topological alignment of single-cell multi-omics integration (UnionCom), Single-Cell alignment using Optimal Transport (SCOT), and Manifold Aligning GAN (MAGAN).

Line 421, Page 11: “In addition to the pairwise distances of cells on the common latent space, we also used the metric, fractions of samples closer than the true match (FOSCTTM)¹⁸ for evaluation. In particular, we calculated the FOSCTTM score of aligned cells as follows. For each cell in the electrophysiological data, we first find its true match in the gene expression data, and then rank all other cells on the aligned latent space based on their distances from x , and then compute the fraction of cells that are closer than the true match.”

Reviewers' comments:

Reviewer #1 (Remarks to the Author):

All my questions have been addressed.

Reviewer #3 (Remarks to the Author):

The authors have addressed many of my comments but there are still some major concerns that remain. In particular, the authors compared their method with UnionCom. Their results seem to indicate that UnionCom performs better than their proposed method on the experiments. So these are my remaining questions:

- 1) If UnionCom outperforms the proposed method, when would it make sense to use the authors' method?
- 2) Are the comparisons with UnionCom run for the purely unsupervised UnionCom or the unsupervised version using the same cross-correspondence as the authors' approach? If the cross-correspondence was not used, I imagine that UnionCom would do even better with it.
- 3) Does UnionCom or MAGAN recover the single trajectory for t-types, and the clusters? Or does only the proposed method recover such structures?
- 4) The comparison with UnionCom with just the boxplots of euclidean distances are insufficient, further experiments as in the original UnionCom paper would be better and more enlightening.